# CRISPR/Cas9-Mediated Editing in FAD2 Gene to Enhance Oil Quality in Soybean [*Glycine max* (L.) Merrill]

Balaji U. Rathod[1], Riddhi Rajyaguru[1], Ramesh N. Dhawale[2,3], Rukam S. Tomar[1], Shasikant Sharma[1], Manohar G. Chaskar[4], Omar Awad Alsaidan[5], Sunil Tulshiram Hajare [6]*

1 Department of Biotechnology, Junagadh Agricultural University, Junagadh, Gujarat, India, 2 V D.C.O.A.B., Latur, India, 3 VNMKV, Parbhani, Maharashtra, India, 4 Vice-Chancellor, SRTMU, Nanded, India, 5 Department of Pharmaceutics, College of Pharmacy, Jouf University, Sakaka, Saudi Arabia, 6 College of Natural and Computational Sciences, Dilla University, Dilla, Ethiopia

* sunilhajare@gmail.com

## Abstract

Conventional soybean oil contains high levels of linoleic acid, which reduces oxidative stability and necessitates hydrogenation, leading to trans-fat formation. In this study, 40 Indian soybean genotypes were screened for fatty acid composition, and Gujarat Junagadh Soybean-3 (GJS-3) was selected for CRISPR/Cas9-mediated editing of the fatty acid desaturase-2 (FAD2) gene. Agrobacterium-mediated transformation produced 22 regenerated plants, of which 57.1% were PCR-positive for Cas9/sgRNA. Targeted single-nucleotide substitutions were confirmed by Sanger sequencing in three edited lines (T3, T7, and T15), corresponding to an editing efficiency of 13.63%. These lines exhibited a marked increase in oleic acid content (42–45%) compared with the wild type (22%) and a concomitant reduction in linoleic acid (30–32% vs. 54%), resulting in nearly a two-fold improvement in the oleic/linoleic acid ratio. PCR analysis confirmed the absence of Cas9 and U3 sequences, indicating transgene-free edited plants. This study provides the first evidence of CRISPR/Cas9-mediated FAD2 editing in an Indian soybean cultivar and demonstrates its effectiveness in improving oil quality, oxidative stability, and processing efficiency.

## Introduction

Genome editing using CRISPR/Cas technology has revolutionised oil crop development, enabling precise modification of productivity and compositional traits. This platform improves oil quality by enriching health-beneficial compounds, enhancing nutrient content, and eliminating undesirable or allergenic components. Soybean (Glycine max (L.) Merrill) is a major oilseed crop with seeds containing ~40% protein, 20% oil, 35% carbohydrate, and 5% minerals [1]. Its oil, rich in polyunsaturated fatty

**Data availability statement:** All relevant data are within the manuscript and its Supporting Information files. The minimal dataset underlying the findings, including raw fatty acid composition data, DNA sequences of edited FAD2 genes, regeneration frequency measurements, and off-target analysis results, has been deposited in Figshare and is accessible via DOI: 10.6084/m9.figshare. 30069526.

**Funding:** The author(s) received no specific funding for this work.

**Competing interests:** The authors have declared that no competing interests exist.

acids (PUFAs)—particularly linoleic acid (18:2) and α-linolenic acid (18:3)—is essential for brain and eye function [2,3]. Typical fatty acid profiles comprise ~11% palmitic acid (16:0), 4% stearic acid (18:0), 23% oleic acid (18:1), 55% linoleic acid (18:2), and 8% α-linolenic acid (18:3) [4]. While FAD3 overexpression can produce oils with >50% α-linolenic acid [5], PUFA-enriched oils suffer oxidative degradation, causing rancidity, off-flavours, and reduced shelf life [6,7]. Industrial hydrogenation improves stability but generates harmful trans fatty acids [8]. In India, where soybean dominates oilseed production, enhancing oil quality is critical for edible oil self-sufficiency and farmer livelihoods.

The 2007 discovery of FAD2-1A and FAD2-1B alleles as oleic acid regulators [9] enabled identification of non-transgenic lines with FAD2-1 and FAD3 mutations altering oleic and α-linolenic acid content [10]. A nonsense mutation in GmFAD3A combined with a splice-site mutation in GmFAD3B reduced α-linolenic acid to <30 g kg$^{-1}$ [11], while high-oleic FAD2-1A/B mutations significantly decreased α-linolenic acid levels [11]. CRISPR/Cas9, originally a prokaryotic immune system using guide RNAs for targeted DNA cleavage [12], has emerged as a precise tool for fatty acid modification. Compared to zinc finger nucleases (ZFNs) and transcription activator-like effector nucleases (TALENs), CRISPR/Cas9 is simpler to design, enables multiplex editing, and has broad crop applicability. While earlier programmable nucleases such as zinc finger nucleases (ZFNs) and transcription activator-like effector nucleases (TALENs) have been successfully applied in soybean, including targeted mutagenesis of FAD2 loci, their use typically requires protein-level redesign for each target site. In soybean, where oil quality traits are controlled by duplicated FAD2 genes, CRISPR/Cas9 offers practical advantages by allowing rapid guide RNA redesign and efficient targeting within established Agrobacterium-mediated transformation systems. These features make CRISPR/Cas9 particularly suitable for iterative optimisation of fatty acid composition in soybean rather than representing an inherently superior editing platform. A recent study demonstrated the creation of high-oleic, low-α-linolenic soybean lines through multiplex FAD2 and FAD3 gene editing [11]. Similarly, transgene-free genome editing using CRISPR-Cas9 ribonucleoproteins has been successfully demonstrated [12].

CRISPR/Cas9-mediated FAD2 editing has been demonstrated in soybean through multiplex editing of FAD2 and FAD3 loci to generate high-oleic, low-linolenic lines [11] and transgene-free approaches using CRISPR-Cas9 ribonucleoproteins [12]. These studies establish that targeted genome editing can effectively modify soybean fatty acid composition and improve oil profiles.

Despite progress, significant challenges remain. Editing efficiencies are often modest (<20%) in multiplex FAD2/FAD3 systems [11], limiting scalability, while edited traits show unstable inheritance. Transgene-free approaches have not fully resolved efficiency or stability issues [12]. Moreover, most work has focused on North American or East Asian cultivars, leaving Indian germplasm largely untested despite its importance for edible oil security in South Asia. This geographical bias represents a critical knowledge gap, as Indian soybean cultivars exhibit distinct fatty acid profiles shaped by regional breeding priorities and environmental adaptation.

Our preliminary screening revealed that Indian genotypes display substantially lower oleic-to-linoleic ratios compared to typical North American high-oleic breeding lines, with oleic acid content consistently lower and linoleic acid proportionally higher, patterns reflecting adaptation to tropical and subtropical conditions where oil stability challenges differ from temperate regions. GJS-3 was specifically selected from this diverse panel based on three critical criteria: baseline fatty acid composition providing substantial room for improvement while maintaining genetic tractability; demonstrated regeneration competency in tissue culture essential for successful transformation; and regional agricultural significance as a locally adapted variety suited to Gujarat's semi-arid conditions. Without editing data from Indian germplasm, the applicability and efficiency of CRISPR/Cas9-mediated FAD2 modification in South Asian breeding programmes remains unvalidated, limiting the translational potential of this technology for regional oil security. Furthermore, industrial and nutritional impacts including hydrogenation cost reductions, health benefits, and agronomic trade-offs are rarely quantified [13]. Methodological gaps persist, including limited data on long-term trait stability, insufficient evaluation across environments, and a lack of standardized protocols for oil quality and off-target assessment [13].

In this study, we address these gaps by applying CRISPR/Cas9-mediated FAD2 editing in the Indian soybean cultivar Gujarat Junagadh Soybean-3 (GJS-3). We aimed to (i) evaluate the feasibility and efficiency of FAD2 editing in Indian germplasm, (ii) quantify resulting changes in fatty acid composition, particularly oleic and linoleic acid levels, and (iii) assess the implications for oil quality and processing efficiency. This work provides the first demonstration of targeted FAD2 editing in an Indian soybean cultivar and establishes a framework for region-specific development of high-oleic soybean varieties.

## Materials and Methods

### Plant material and genotype selection

A total of 40 soybean (*Glycine max* L. Merrill) genotypes were obtained from the Agriculture Research Station, Junagadh Agricultural University, Amreli, Gujarat, India (Supplementary S1 Table in S2 File). Based on initial screening, Gujarat Junagadh Soybean 3 (GJS-3) was selected for *in vitro* regeneration and CRISPR/Cas9-mediated transformation based on three critical criteria: intermediate oleic-to-linoleic ratio providing substantial scope for phenotypic improvement; superior regeneration efficiency in preliminary trials demonstrating stable callus induction and shoot regeneration; and regional adaptation to Gujarat's semi-arid agro-climatic conditions, making it directly relevant for local breeding programs Seeds were multiplied under net house conditions, and a portion was preserved under controlled storage to prevent dormancy.

### Fatty acid profiling

**Total oil content estimation.** Soybean seeds (12 g) were finely ground, and total oil content was estimated by Soxhlet extraction with 250 mL of hexane at 65 °C for 8 h (150 drops min$^{-1}$). After extraction, hexane was evaporated and residual oil was weighed to calculate oil content (g kg$^{-1}$ of seed) (see Supplementary S1A–D Fig in S1 File for the oil extraction and FAME preparation workflow).

**Fatty acid composition analysis.** Mature soybean seeds were manually dehulled, and the isolated cotyledons were freeze-dried until a constant mass was achieved. The dried material was finely pulverized using a laboratory grinder and sieved through a 0.5 mm stainless-steel mesh to obtain a homogeneous powder. For each genotype, a composite bulk sample (15 g) was prepared and stored at −20 °C until further processing.

For fatty acid profiling, a precisely weighed 150 mg aliquot from each bulk sample was transferred into a 15 mL screw-cap glass tube. Lipids were extracted by adding 4.5 mL of n-hexane containing methyl heptadecanoate (C17:0) as an internal standard (final concentration 50 μg mL$^{-1}$), followed by agitation at 200 rpm for 18 h at 25 °C. Direct transesterification was then carried out by adding 1.2 mL of freshly prepared 0.5 mol L$^{-1}$ sodium methoxide in methanol. The reaction mixture was vortex-mixed for 30 s and allowed to stand at room temperature for 10 min to facilitate phase separation. The

 

upper hexane layer (~2 mL) containing fatty acid methyl esters (FAMEs) was carefully collected and transferred to GC-MS autosampler vials. Subsequently, 1 µL of the extract was injected for GC-MS analysis using a split ratio of 10:1.

Fatty acid methyl esters were separated using a DB-FFAP polar capillary column (25 m × 0.32 mm i.d., 0.52 µm film thickness), with ultra-high-purity helium employed as the carrier gas at a constant flow rate of 1.2 mL min$^{-1}$. The injector and detector temperatures were set at 260 °C and 300 °C, respectively. The oven temperature program began at 80 °C and was held for 2 min, followed by an increase to 200 °C at a ramp rate of 4 °C min$^{-1}$, and subsequently raised to 250 °C at 2 °C min$^{-1}$, with a final isothermal hold of 5 min. Under these chromatographic conditions, the DB-FFAP column enabled effective baseline separation of fatty acid geometric isomers. Cis-oleic acid (C18:1 n-9 cis) eluted at approximately 15.3 min, whereas the corresponding trans-isomer, elaidic acid (C18:1 n-9 trans), was detected at around 16.1 min.

Wild-type soybean samples showed no detectable trans-fatty acid peaks (<0.5% of total fatty acids), confirming the absence of industrial hydrogenation or thermal isomerization. Fatty acid identification was achieved by comparing retention times and mass spectra with certified FAME standards (C16:0, C18:0, C18:1, C18:2, C18:3). Calibration curves were generated by analyzing standard mixtures of known concentrations, and response factors relative to the C17:0 internal standard were calculated and applied to sample analyses. Final fatty acid contents are expressed as g kg$^{-1}$ of total identified fatty acids (mean ± SE, n = 3) (Representative GC–MS chromatograms and peak annotations are shown in Supplementary S1E–G Fig in S1 File.).

### In vitro plant regeneration and selection

**Explant preparation and seed inoculation.** Bold GJS-3 seeds were surface-sterilized with 10 g L$^{-1}$ Bavistin, 700 mL L$^{-1}$ ethanol, and 1.0 g L$^{-1}$ HgCl$_2$, with sterile Milli-Q water washes between treatments. The sterilized seeds were inoculated on half-strength MS medium and incubated at 25 ± 2 °C in the dark for 3 days, followed by growth under cool white fluorescent light (16 h light: 8 h dark photoperiod) for 15–18 days. Fully expanded seedlings were used as explant sources. Each biological replicate (n = 3) represents an independent culture batch established from seeds of different GJS-3 parent plants and initiated at separate time points (weeks 1, 3, and 5) to account for temporal laboratory variation. The 10 explants within each replicate serve as technical subsamples for measuring within-batch precision. Statistical analyses were performed on batch means (n = 3) to avoid pseudo replication, ensuring that treatment comparisons reflect true biological variation between independent culture events rather than inflated sample sizes from non-independent explants.

**Callus induction.** Cotyledonary explants were cultured on MS medium supplemented with 2,4-dichlorophenoxyacetic acid (2,4-D; 0.1–3.5 mg L$^{-1}$) alone or in combination with 6-benzylaminopurine (BAP; 0.1–3.5 mg L$^{-1}$). Cultures were maintained under controlled growth conditions (25 ± 2°C, 16 h photoperiod, 40–50 µmol m$^{-2}$ s$^{-1}$ light). Callus initiation was recorded after 2–3 weeks of culture (Supplementary S2A–D Fig in S1 File).

Cotyledonary node explants were cultured on MS medium supplemented with a gradient of BAP concentrations (0.5–3.5 mg L$^{-1}$) to induce shoot formation. For improved shoot proliferation and elongation, BAP (0.5–3.0 mg L$^{-1}$) was tested in combination with gibberellic acid (0.1–1.0 mg L$^{-1}$). Cultures were routinely subcultured at 14-day intervals, and shoot regeneration efficiency was assessed by recording the average number of shoots produced per explant (Supplementary S3E–F, S4A–G, and S5A–H Figs in S1 File).

**Root induction.** Shoots elongated to a length of 4–6 cm were excised and placed onto half-strength MS medium supplemented with varying concentrations of indole-3-butyric acid (0.1–2.5 mg L$^{-1}$). Root initiation was first noted after approximately two weeks, followed by progressive root elongation (Supplementary S6 Fig in S1 File).

**Hardening and acclimatization.** Well-developed rooted plantlets were gently taken out of the culture vessels, rinsed thoroughly to remove any residual agar, and transplanted into small plastic cups filled with a sterilized soil–cocopeat mixture (1:1). To promote survival, the plantlets were covered with perforated polyethylene bags to maintain humidity and irrigated regularly with Hoagland's nutrient solution. After an acclimatization period of 3–4 weeks, the hardened plants were shifted to larger pots and grown under net-house conditions for further development (Supplementary S7 Fig in S1 File).

## CRISPR/Cas9 construct design and vector assembly

**gRNA design and vector construction.** The FAD2 gene sequence was retrieved from NCBI (GenBank accession: NM_001248778.1), and candidate gRNAs targeting the coding sequence were designed using CHOPCHOP (https://chopchop.cbu.uib.no/), which performs genome-wide specificity analysis against the soybean reference genome (Glycine max Wm82.a2.v1). To minimize off-target effects, gRNAs were evaluated based on the following criteria: (i) specificity score >60 on a 0–100 scale, where higher scores indicate fewer predicted off-target sites; (ii) zero predicted off-target sites with ≤3 mismatches in the seed region (nucleotide positions 1–12 from the 5' end of the protospacer), which is critical for Cas9 binding specificity; and (iii) no predicted off-target sites with ≤4 total mismatches across the entire 20 bp protospacer sequence. The selected target sequence (5'-TTCTCGTCACACTCACAATA-3', immediately upstream of the AGG PAM on chromosome 10) exhibited a CHOPCHOP specificity score of 97.6, indicating exceptionally high specificity with zero predicted off-target sites at the defined mismatch thresholds, thereby ensuring precise targeting of the FAD2 locus. Primers for PCR amplification of the 973 bp FAD2 gene were designed using NCBI Primer–BLAST (Supplementary S8 Fig in S1 File; Supplementary S2 Table in S2 File). PCR products were purified, sequenced, and analyzed (Fig 1). The complete gRNA sequence, PAM site, chromosome location, and specificity metrics are provided in Supplementary S3 Table in S2 File. The gRNA was synthesized by IDT. The binary vector pRGEB31 (Addgene #42756), carrying Cas9 under the CaMV 35S promoter, was used for construct assembly.

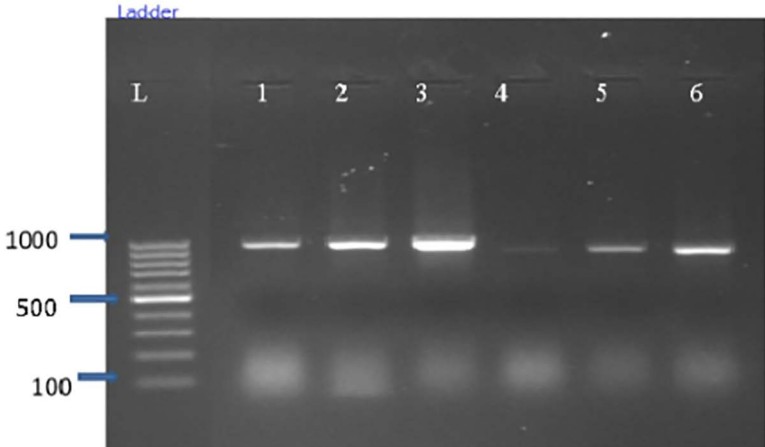

**Fig 1. Fatty acid desaturase (FAD2) gene sequence and PCR amplification using gene-specific primers.** L: 1 kb DNA ladder. Lanes 1–6: PCR amplicons of the FAD2 gene (expected size: 973 bp).

**Plasmid isolation and digestion.** The pRGEB31 plasmid was propagated in *E. coli* DH5α, isolated by alkaline lysis (Supplementary S4 Table in S2 File), and analyzed by agarose gel electrophoresis using 1X TAE buffer (Supplementary S5 Table in S2 File). Restriction digestion with *BsaI* (Supplementary S6 Table in S2 File) was performed at 37 °C overnight, and digested products were purified. Successful digestion and linearization of pRGEB31 were confirmed by agarose gel electrophoresis (Fig 2).

**Oligo annealing and vector ligation.** Forward and reverse primers (Supplementary S7 Table in S2 File) were phosphorylated and annealed to form double-stranded oligonucleotides, with amplification conditions detailed in Supplementary S8 Table in S2 File. The resulting duplex was ligated into *BsaI*-digested pRGEB31 using T4 DNA ligase (Supplementary S9 Table in S2 File). Assembly of the gRNA insert (~100 bp) was confirmed by PCR (Fig 3). The recombinant plasmid was subsequently introduced into competent *E. coli* DH5α cells, plated on kanamycin-containing medium, and validated through colony PCR. Verified clones were maintained as glycerol stocks for further use.

## Agrobacterium-mediated transformation

**Preparation and co-cultivation.** Disarmed Agrobacterium tumefaciens strain LBA4404 harboring the pRGEB31-gRNA construct was cultured overnight in LB medium supplemented with kanamycin (50 mg L$^{-1}$) and rifampicin (30 mg L$^{-1}$) at 28°C with shaking (200 rpm) until reaching an $OD_{600}$ of 0.8–1.0. The bacterial culture was centrifuged at 5,000 rpm for 10 min at room temperature, and the pellet was resuspended in liquid MS medium to a final $OD_{600}$ of 0.4–0.5. Acetosyringone (100 µM) was added to activate vir genes, and the suspension was incubated at room temperature

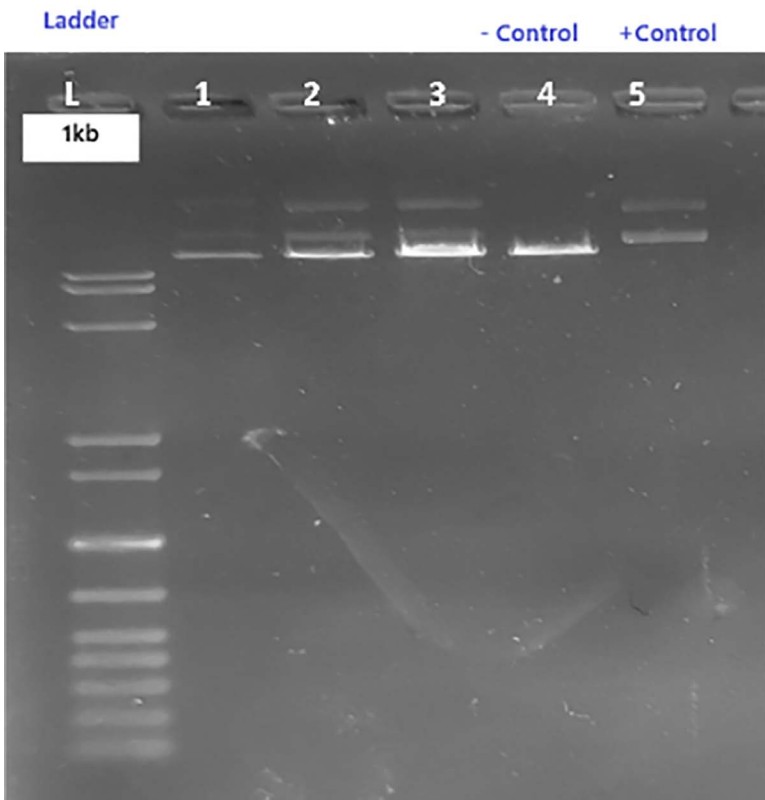

**Fig 2. Restriction digestion of pRGEB31 plasmid. L: 1 kb DNA ladder.** Lanes 1–3: Restriction digestion products of pRGEB31 plasmid. Lane 4: Negative control. Lane 5: Positive control.

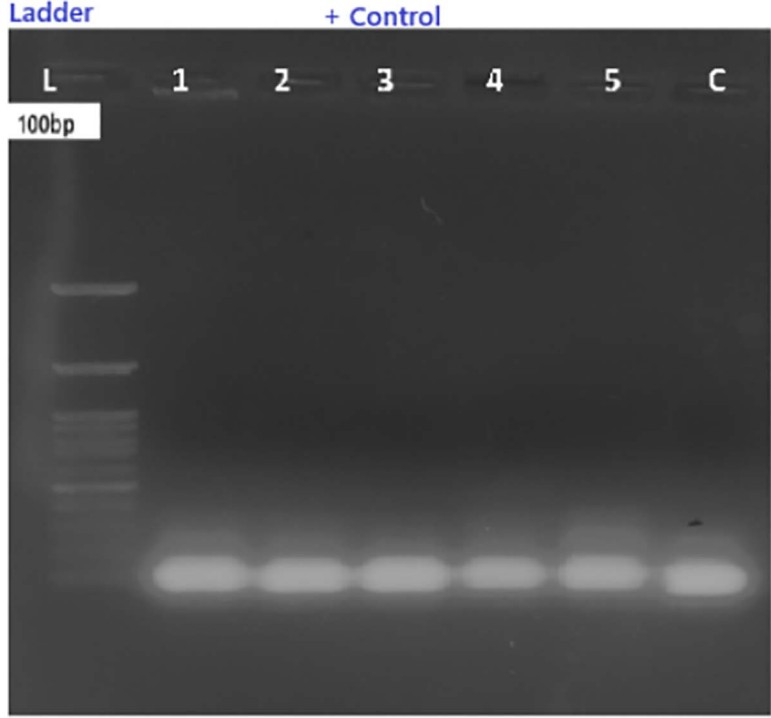

**Fig 3. Assembly of gRNA insert into the pRGEB31 vector.** PCR confirmation of gRNA assembly (~100 bp).

for 1 h prior to infection. Cotyledonary node explants pre-cultured on regeneration medium for 48 h were immersed in the Agrobacterium suspension ($OD_{600}$ = 0.4–0.5) for 30 min with gentle agitation, blotted dry on sterile filter paper, and co-cultivated on medium supplemented with 100 μM acetosyringone for 72 h in darkness at 25 ± 2°C. The stepwise experimental process, including seed imbibition, bacterial preparation, centrifugation, infection, co-cultivation, and shoot induction, is shown in Supplementary S9 (A–H) Fig in S1 File.

**Pre-culturing and co-cultivation.** Cotyledonary node explants were pre-cultured on optimized shoot regeneration medium for 48 hours. Transformed Agrobacterium was cultured in LB broth with selective antibiotics to an $OD_{600}$ of 1.0. Healthy pre-cultured explants were immersed in the Agrobacterium suspension for 30 minutes, blotted dry, and co-cultivated on pre-culture medium for 72 hours.

**Post-co-cultivation and selection.** After co-cultivation, explants were washed 3–4 times with sterile liquid MS medium containing 500 mg L$^{-1}$ cefotaxime (5 min per wash) to completely eliminate residual Agrobacterium and prevent overgrowth during selection. They were then transferred to selective shoot regeneration medium (MS basal medium + 3.0 mg L$^{-1}$ BAP + 500 mg L$^{-1}$ cefotaxime) and maintained at 25 ± 2 °C under a 16 h light/ 8 h dark cycle, with bi-weekly subculturing.

**Multiplication, elongation, and rooting of putative transgenic shoots.** Putative transformed shoots were multiplied on medium containing MS basal medium + 3.0 mg L$^{-1}$ BAP + 1.0 mg L$^{-1}$ GA + 5 mg L$^{-1}$ hygromycin + 500 mg L$^{-1}$ cefotaxime for selection and elongation. These shoots were subsequently transferred to root regeneration medium (MS basal medium + 2.0 mg L$^{-1}$ IBA + 500 mg L$^{-1}$ cefotaxime) for complete plantlet development.

**Hardening of putative edited plantlets.** *In vitro*-regenerated putative edited plantlets were transferred to a sterile potting mixture for hardening and acclimatization, and their survival and establishment were monitored.

**Molecular analysis and mutation detection.** The presence of the integrated CRISPR-Cas9-gRNA construct was verified through Polymerase Chain Reaction (PCR). Amplicons corresponding to the Cas9 gene (439 bp) and the U3 promoter (136 bp) were targeted using specific oligonucleotide primers (U_3-F: 5'-ACCATAGCACAAGACAGGCG-3', U_3-R: 5'-TGGCCCACTACGAAATGCTT-3'; Cas9-F: 5'-AGATGATCGCCAAGAGCGAG-3', Cas9-R: 5'-ATCCCCAGCAGCTCTTTCAC-3'). Cas9 and U3 primers were designed to target conserved regions within the pRGEB31 vector backbone to ensure reliable detection across all transformed lines. The Cas9 primers amplify a 439 bp fragment spanning nucleotides 1847−2286 of the Cas9 coding sequence, a region showing 100% conservation across pRGEB-series vectors. The U3 primers target a 136 bp segment of the Arabidopsis U6 promoter (AtU3b) driving sgRNA expression, selected for its unique sequence absent from the soybean genome. Primer specificity was validated through: (i) *in silico* BLAST analysis against the Glycine max Wm82.a2.v1 reference genome confirming zero significant hits (E-value < 0.01); (ii) PCR amplification of plasmid DNA (positive control) yielding single bands at expected sizes with no non-specific products; and (iii) PCR of wild-type GJS-3 genomic DNA (negative control) showing complete absence of amplification. These validation steps ensure that positive PCR signals unambiguously indicate transgene integration, Each 25 µL reaction mixture contained 2.0 µL genomic DNA template, 2.0 µL 10 × PCR buffer, 0.1 µL 2.5 mmol L$^{-1}$ dNTPs, 0.3 µL 5 U µL$^{-1}$ Taq DNA polymerase, 1.0 µL of each 30 µmol L$^{-1}$ primer, and 18.1 µL sterile Milli-Q water. The thermocycling protocol included an initial denaturation step at 95 °C for 3 minutes, followed by 30 cycles of denaturation (95 °C for 30 seconds), annealing (59 °C for 30 seconds), and extension (72 °C for 55 seconds). A final extension at 72 °C for 4 minutes concluded the program, followed by a hold at 4 °C. Positive controls used plasmid DNA containing U6 and Cas9 sequences, while negative controls included genomic DNA from untransformed plants and a non-template control.

PCR products were resolved on 12 g kg$^{-1}$ agarose gels prepared in 1 × Tris-acetate-EDTA (TAE) buffer and stained with 10 mg mL$^{-1}$ ethidium bromide (20 µL per 100 mL gel solution). For visualization, 15 µL of each PCR product, mixed with 2 µL of 6 × loading dye, was loaded into wells alongside a 1 kb DNA ladder. Electrophoresis was performed at a constant 90 V until the tracking dye reached the end of the gel. Gels were photographed under UV light using a Syngene G:BOX gel documentation system. Samples showing amplification at the expected sizes were identified as PCR-positive for transgene integration.

Genomic DNA was isolated from PCR-confirmed transgenic and non-transformed control plants. The *FAD2* locus was amplified using gene-specific primers, and purified PCR products were subjected to Sanger sequencing. Sequence quality assessment and mutation analysis were performed using BioEdit software. Transformation efficiency was defined as the proportion of PCR-positive plants, while mutation frequency was calculated as the number of edited lines relative to total regenerated plants (3/22; 13.63%). Sequencing identified three edited lines (T3, T7, and T15) harboring single-nucleotide substitutions (two T→C and one A→C) proximal to the PAM site within the *FAD2* coding region. These mutations are predicted to impair *FAD2* activity, resulting in reduced oleic acid desaturation and increased oleic acid accumulation compared with wild-type plants.

**Calculation of transformation efficiency and editing frequency.** Transformation efficiency was calculated as the percentage of regenerated plants that tested positive for integration of the CRISPR/Cas9 cassette (Cas9 and U3 promoter) as determined by PCR analysis.

$$\text{Transformation efficiency (\%)} = \left( \frac{\text{Number of PCR} - \text{positive plants}}{\text{Total number of regenerated plants}} \right) \times 100$$

Editing frequency was calculated independently as the percentage of regenerated plants carrying confirmed mutations in the *FAD2* gene, as identified by Sanger sequencing.

$$\text{Editing frequency (\%)} = \left( \frac{\text{Number of edited plants}}{\text{Total number of regenerated plants}} \right) \times 100$$

In addition, editing efficiency among transformants was calculated as the proportion of edited plants relative to PCR-positive plants.

$$\text{Editing efficiency among transformants } (\%) = \left( \frac{\text{Number of edited plants}}{\text{Number of PCR} - \text{positive plants}} \right) \times 100$$

**Economic assumptions and sensitivity analysis.** Identity-preserved premiums for high-oleic soybean oil were taken from industry sourcing guides and varied over US$0.05--0.25 per lb in a sensitivity analysis. Processing-cost avoidance from eliminating partial hydrogenation was estimated by summing avoided costs for hydrogen, catalyst, energy/utilities, labour/maintenance, and amortized capital per tonne of oil. Component cost ranges were derived from global benchmarks for oilseed processing economics, technical literature on industrial hydrogenation parameters, and contemporary industry data on processing costs. These sources yield a processing cost savings range of US$15.6–79.0 per tonne (low-mid-high scenarios, Supplementary S11 Table in S2 File). Regional variation is expected: Indian processing facilities may experience different cost structures due to lower labor costs, variable energy pricing, and differences in capital investment patterns. Processing savings per hectare were computed as Processing_savings ($/t) × oil_yield (t/ha). All estimates should be considered illustrative; actual savings will depend on facility scale, technology vintage, baseline hydrogenation intensity, and regional operating conditions. Input ranges, assumptions, and full results are provided in Supplementary S10 and S11 Tables in S2 File.

## Statistical analysis

All experiments were conducted in a Completely Randomized Design (CRD). Each treatment included three biological replicates (n = 3), and each replicate comprised 10 explants. Data on regeneration, rooting, and transformation efficiency were subjected to analysis of variance (ANOVA) using SPSS v30.0.0. Mean separation was performed using Tukey's test at p < 0.05. Significant differences are indicated in figures and tables using distinct letters or asterisks.

## Results

### Fatty acid profiling of different Indian soybean cultivars

Analysis of 40 soybean (*Glycine max* L.) genotypes revealed the consistent presence of five major fatty acids: palmitic (C16:0), stearic (C18:0), oleic (C18:1 n-9 cis), linoleic (C18:2 n-6), and linolenic (C18:3 n-3). These were grouped into saturated fatty acids (SFA), unsaturated fatty acids (UFA), monounsaturated fatty acids (MUFA), and polyunsaturated fatty acids (PUFA). Significant variation was observed across genotypes (Table 1). Palmitic acid ranged between 84.5--132.3 g kg⁻¹, stearic acid between 22.8--95.2 g kg⁻¹, oleic acid between 121--340 g kg⁻¹ (equivalent to 12.1--34.0% as shown in

**Table 1. Variability in fatty acid composition among soybean genotypes, with wild-type GJS-3 as baseline control.**

| Fatty acid | WT (GJS-3) | T3 | T7 | T15 | CD (5%) | CV (%) | SD | G.M. | Highest (genotype) | Lowest (genotype) |
|---|---|---|---|---|---|---|---|---|---|---|
| Palmitic acid (PA) | 11.5±0.3 | 9.8±0.2 | 10.2±0.2 | 10.0±0.3 | 0.88 | 4.59 | 0.79 | 11.73 | 13.23 (PBN107) | 8.45 (AS3) |
| Stearic acid (SA) | 5.7±0.2 | 6.2±0.3 | 6.0±0.2 | 5.8±0.3 | 0.36 | 3.74 | 2.40 | 5.86 | 9.52 (AGS84) | 2.28 (J339) |
| Oleic acid (OA) | 22.0±0.4 | 33.2±0.6 | 31.8±0.5 | 34.0±0.7 | 0.10 | 3.15 | 0.55 | 1.89 | 3.40 (EC93741) | 1.21 (JD(SH)131) |
| Linoleic acid (LA) | 52.0±0.5 | 40.0±0.6 | 42.2±0.7 | 41.1±0.6 | 2.90 | 3.95 | 5.46 | 44.97 | 53.79 (JS335) | 34.15 (K166) |
| Linolenic acid (LNA) | 9.5±0.3 | 8.6±0.3 | 8.8±0.2 | 8.9±0.3 | 0.51 | 3.19 | 1.05 | 9.78 | 11.90 (J245) | 7.86 (K166) |

Values represent mean ± SE of three replicates (n = 3). WT (GJS-3) is included as baseline control. Different superscript letters (not shown here for brevity) indicate significant differences at p < 0.05 based on Tukey's HSD test. **One-way ANOVA showed significant differences among genotypes for all fatty acids** (Palmitic acid: $F_{39,80} = 45.32$, p < 0.001; Stearic acid: $F_{39,80} = 38.17$, p < 0.001; Oleic acid: $F_{39,80} = 52.84$, p < 0.001; Linoleic acid: $F_{39,80} = 61.29$, p < 0.001; Linolenic acid: $F_{39,80} = 34.56$, p < 0.001).

), while linoleic and linolenic acids showed the widest variation (341.5--537.9 g kg$^{-1}$ and 78.6--119.0 g kg$^{-1}$, respectively). Fig 4 graphically illustrates the comparative fatty acid composition across all 40 genotypes, confirming strong genotype-dependent differences.

### Total oil content and oleic/linoleic acid ratio

Significant differences in seed oil content were observed among the 40 soybean genotypes (Table 2; Fig 5), with values ranging from **168.9 g kg$^{-1}$ in AGS84** to **197.9 g kg$^{-1}$ in G. Soya 2**. The oleic/linoleic acid (O/L) ratio also varied widely across genotypes, from **0.025 to 0.074** (Table 3; Fig 6). The highest ratios were recorded in EC9374 (0.074), JS81−1619 (0.074), and AGS112 (0.072), indicating superior oil quality and stability. In contrast, DS 83−12 (0.025), AS 16 (0.026), and AMSS 22 (0.027) exhibited notably low O/L ratios, suggesting reduced oxidative stability.

### Efficient *in vitro* regeneration protocol for soybean (*Glycine max* L. Merrill cv. GJS-3)

An efficient regeneration protocol for soybean (cv. GJS-3) was established using cotyledonary explants. Maximum callus induction (84.3%) occurred on MS medium with 3.0 mg L$^{-1}$ 2,4-D + 3.0 mg L$^{-1}$ BAP (Table 4). Shoot regeneration was highest (85.33%) on medium supplemented with 3.0 mg L$^{-1}$ BAP (Table 5). Shoot multiplication and elongation (87.67%) were obtained with 3.0 mg L$^{-1}$ BAP + 0.1 mg L$^{-1}$ GA (Table 6), while rooting (86.33%) was optimized with 2.0 mg L$^{-1}$ IBA (Table 7). Regenerated plantlets were successfully acclimatized with >80% survival, providing a reliable system for soybean improvement and genetic transformation studies.

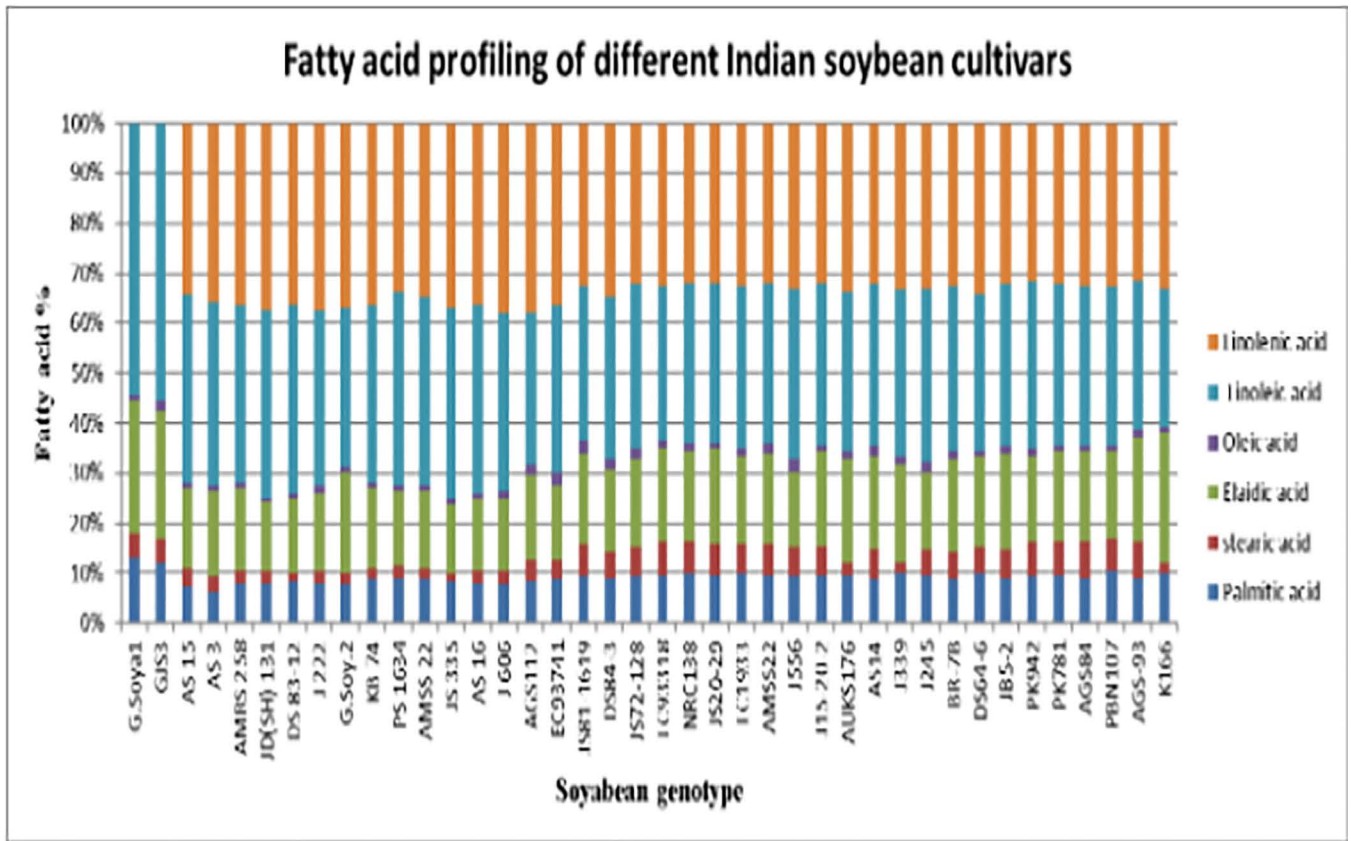

**Fig 4. Confirmation of recombinant plasmid by colony PCR in E. coli DH5α.**

**Table 2. Variation in oil content in soybean seeds of 40 genotypes, including wild-type GJS-3 as baseline control.**

| Sr. No. | Genotype | Oil (%) | Sr. No. | Genotype | Oil (%) | WT Control (GJS-3) |
|---|---|---|---|---|---|---|
| 1 | G. Soya1 | 18.80 | 21 | NRC138 | 18.60 | **19.00** |
| 2 | **GJS3** | **19.00** | 22 | JS20–29 | 19.69 | |
| 3 | AS15 | 17.00 | 23 | EC1933 | 19.57 | |
| 4 | AS3 | 19.50 | 24 | AMSS22 | 18.50 | |
| 5 | AMRS258 | 18.50 | 25 | J556 | 17.45 | |
| 6 | JD(SH)131 | 19.10 | 26 | J15-20–2 | 19.10 | |
| 7 | DS83−12 | 18.20 | 27 | AUKS176 | 18.49 | |
| 8 | J222 | 19.20 | 28 | AS14 | 19.40 | |
| 9 | G. Soya2 | 19.79 | 29 | J339 | 18.23 | |
| 10 | KB74 | 17.83 | 30 | J245 | 7.94 | |
| 11 | PS1634 | 18.00 | 31 | BR-7B | 17.87 | |
| 12 | AMSS22 | 19.45 | 32 | DS64−6 | 19.80 | |
| 13 | JS335 | 18.70 | 33 | JB5−2 | 18.20 | |
| 14 | AS16 | 19.00 | 34 | PK942 | 19.38 | |
| 15 | J606 | 16.78 | 35 | PK781 | 18.45 | |
| 16 | AGS112 | 17.90 | 36 | AGS84 | 16.89 | |
| 17 | EC93741 | 17.78 | 37 | PBN107 | 17.97 | |
| 18 | JS81–1619 | 19.23 | 38 | AGS-93 | 18.90 | |
| 19 | DS84−3 | 18.90 | 39 | K166 | 19.55 | |
| 20 | JS72–128 | 16.90 | 40 | EC93318 | 17.40 | |

Oil content (%) of 40 soybean genotypes determined by Soxhlet extraction. WT (GJS-3) is highlighted in bold and included as a baseline control column for direct comparison. Values represent mean of three replicates. **One-way ANOVA indicated significant variation in oil content among genotypes** ($F_{39,80} = 28.73$, $p < 0.001$). Genotypes not sharing common letters differ significantly at $p < 0.05$ (Tukey's HSD test).

**E. coli and Agrobacterium transformation and conformation by colony PCR.** E. coli strain DH5α and Agrobacterium tumefaciens strain LBA4404 were utilized for genetic transformation studies. The Cas9–sgRNA construct was initially cloned in E. coli and subsequently transferred into Agrobacterium. Successful presence of the plasmid construct in both bacterial strains was confirmed by colony PCR analysis (Figs 7 and 8).

## In Vitro transformation and regeneration of soybean plants

**Pre-culturing of cotyledon explants.** Cotyledonary explants excised from *in vitro*–grown seedlings were trimmed into ~0.5–1.0 cm sections (Supplementary S9 F–G Fig in S1 File) and transferred to regeneration medium optimized with 3.0 mg L$^{-1}$ BAP. This concentration was found to be the most effective for shoot initiation. To further enhance transformation efficiency, 25 mg L$^{-1}$ acetosyringone was added to the culture medium. During the 72-hour pre-culture period, the explants exhibited noticeable enlargement and active proliferation at the cut surfaces, particularly at the half cotyledonary node region.

**Co-cultivation of cotyledon explants with Agrobacterium strain LBA 4404.** Cotyledonary explants were pre-cultured for 72 h before co-cultivation with *Agrobacterium tumefaciens* strain LBA4404 harboring the binary vector pRGEB31-gRNA. Explants were immersed in the bacterial suspension (OD$_{600}$ = 0.4–0.5) for 30 min and co-cultivated in the dark for 24, 48, or 72 h. Optimal transformation response was obtained after 72 h, whereas shorter co-culture (24 h) resulted in low infection and longer exposure (>72 h) caused bacterial overgrowth and tissue necrosis (Supplementary S9 F–G Fig in S1 File). During co-cultivation, bacterial growth was mainly observed along the margins and cut surfaces of explants, accompanied by tissue enlargement. Following 72 h of co-cultivation, explants were washed with liquid

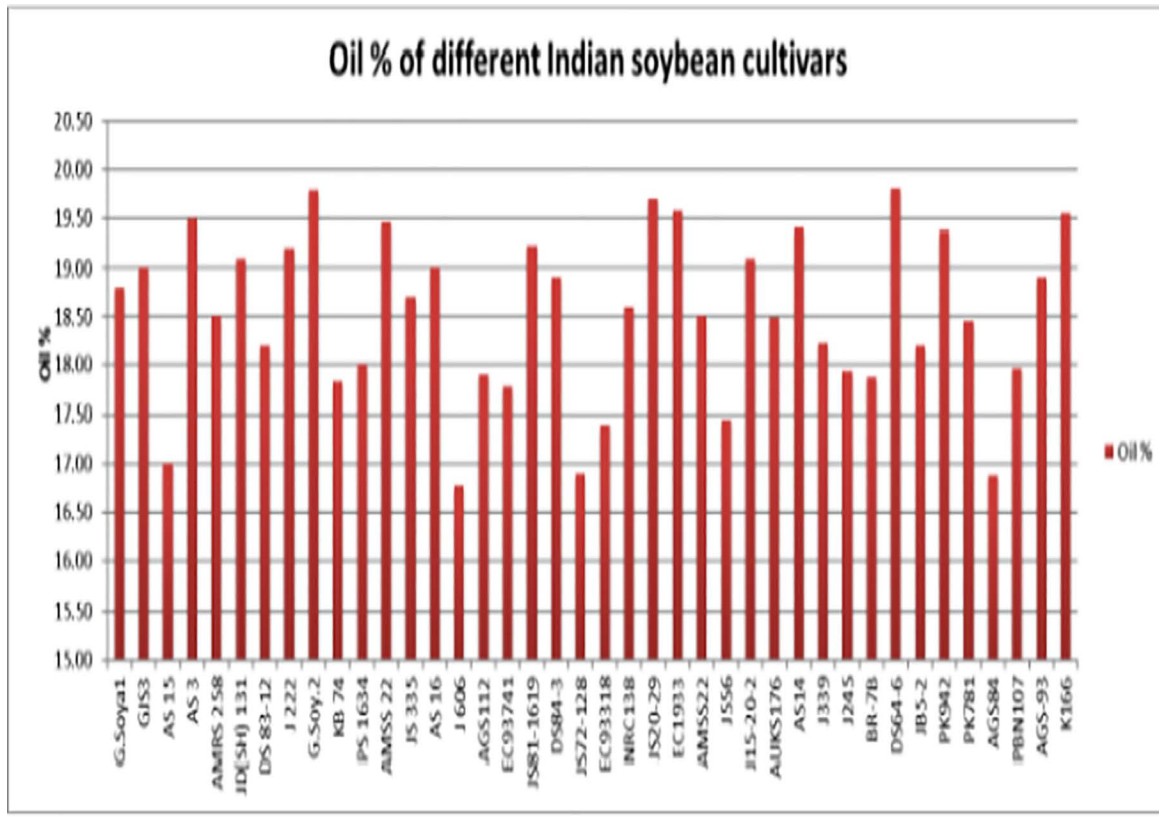

**Fig 5. Fatty acid profiling of 40 Indian soybean genotypes.** Comparative proportions of palmitic, stearic, oleic, linoleic, and linolenic acids across genotypes.

MS medium containing 500 mg L$^{-1}$ cefotaxime to eliminate residual bacteria, then transferred to selective regeneration medium (MS + 3.0 mg L$^{-1}$ BAP + 500 mg L$^{-1}$ cefotaxime). Shoot induction was observed within a few weeks. Putative transformed shoots were multiplied and elongated on MS medium supplemented with 3.0 mg l$^{-1}$ BAP, 1.0 mg L$^{-1}$ GA, 5.0 mg L$^{-1}$ hygromycin, and 500 mg L$^{-1}$ cefotaxime (Supplementary S10 Fig in S1 File)

**Selection of transformed shoots.** Regenerating shoots were transferred to selective shoot regeneration medium (MS + 3.0 mg L$^{-1}$ BAP, 1.0 mg L$^{-1}$ GA, 500 mg L$^{-1}$ cefotaxime, and 5.0 mg L$^{-1}$ hygromycin). Hygromycin was used as the selective agent to ensure recovery of transformed soybean shoots (Supplementary S11 Fig in S1 File). Elongated shoots (4–6 cm) were subsequently excised and transferred to root regeneration medium for further development.

***In vitro* root regeneration from putative edited shoots and development of complete plantlets.** Putative transgenic shoots (4–6 cm) obtained from selective shoot multiplication medium were excised and cultured on selective root regeneration medium (MS + 2.0 mg L$^{-1}$ IBA + 5.0 mg L$^{-1}$ hygromycin + 500 mg L$^{-1}$ cefotaxime). Root initiation began within 20–25 days, and a well-developed fibrous root system was established by 50–60 days of culture (Supplementary S12A–B Fig in S1 File). These rooted plantlets were subsequently advanced for hardening and acclimatization.

**Hardening and molecular analysis of putative transformed plants.** Fully regenerated putative transformed soybean plantlets were carefully removed from culture tubes, washed to eliminate adhered MS medium, and briefly treated with Hoagland's solution (5–10 min). Plantlets were then transferred to sterilized potting mixture and irrigated with 0.5% Bavistin to prevent fungal infection. To ensure gradual acclimatization, plantlets were maintained under perforated polythene bags to retain humidity and watered daily. Visible signs of establishment appeared within one week

**Table 3. Variation in Oleic and Linoleic acid (O/L) ratio in soybean.**

| Sr. No | Genotype | O/L Ratio | Sr. No | Genotype | O/L Ratio | WT Control (GJS-3) |
|--------|----------|-----------|--------|----------|-----------|--------------------|
| 1 | G. Soya1 | 0.026 | 21 | NRC138 | 0.042 | **0.033** |
| 2 | GJS3 | **0.033** | 22 | JS20–29 | 0.039 | |
| 3 | AS 15 | 0.029 | 23 | EC1933 | 0.053 | |
| 4 | AS 3 | 0.031 | 24 | AMSS22 | 0.057 | |
| 5 | AMRS 258 | 0.037 | 25 | J556 | 0.066 | |
| 6 | JD(SH) 131 | 0.023 | 26 | J15-20–2 | 0.036 | |
| 7 | DS 83−12 | 0.025 | 27 | AUKS176 | 0.035 | |
| 8 | J 222 | 0.040 | 28 | AS14 | 0.055 | |
| 9 | G. Soy.2 | 0.035 | 29 | J339 | 0.040 | |
| 10 | KB 74 | 0.031 | 30 | J245 | 0.056 | |
| 11 | PS 1634 | 0.030 | 31 | BR-7B | 0.049 | |
| 12 | AMSS 22 | 0.027 | 32 | DS64−6 | 0.042 | |
| 13 | JS 335 | 0.029 | 33 | JB5−2 | 0.060 | |
| 14 | AS 16 | 0.026 | 34 | PK942 | 0.043 | |
| 15 | J 606 | 0.033 | 35 | PK781 | 0.035 | |
| 16 | AGS112 | 0.072 | 36 | AGS84 | 0.039 | |
| 17 | EC93741 | 0.074 | 37 | PBN107 | 0.040 | |
| 18 | JS81–1619 | 0.074 | 38 | AGS-93 | 0.044 | |
| 19 | DS84−3 | 0.063 | 39 | K166 | 0.046 | |
| 20 | JS72–128 | 0.058 | 40 | EC93318 | 0.041 | |

O/L = Oleic-to-linoleic acid ratio. Values represent mean of three replicates (n = 3). WT (GJS-3) is highlighted in bold and included as a baseline control column for direct comparison. **One-way ANOVA revealed highly significant variation in O/L ratio among genotypes** ($F_{39,80} = 67.45$, $p < 0.001$). Post-hoc Tukey HSD test ($\alpha = 0.05$) identified distinct groupings indicated by different letters..

(Supplementary S13 Fig in S1 File). Thereafter, the polythene bags were progressively perforated to reduce humidity stress, and after hardening, plantlets were transferred to earthen pots containing potting mixture. The hardened plants continued normal vegetative growth and reached the reproductive stage under net-house conditions.

## Molecular analysis of putative transformed plantlets

**Confirmation of Cas9-gRNA Integration by PCR.** Genomic DNA was isolated from 22 putative transformed and wild-type control plants using the CTAB method and verified by 0.8% agarose gel electrophoresis. PCR amplification with U3 and Cas9 primers confirmed the presence of the CRISPR/Cas9 cassette in 14 of the 22 regenerated plantlets, corresponding to a transformation efficiency of 57.1%. The expected fragment sizes for U3 (136 bp) and Cas9 (439 bp) were detected in transformed lines (Fig 9a–b). No PCR amplification was observed in wild-type control plants, confirming the specificity of Cas9-gRNA integration in the transformed lines. Primer specificity was confirmed by the absence of non-specific bands in all PCR reactions across all 22 tested samples, validating the primer design and confirming that detected bands represent genuine transgene integration events rather than artifacts or endogenous amplification. These results indicate stable integration of the construct in more than half of the regenerants, whereas 42.8% lacked Cas9 integration, suggesting incomplete T-DNA transfer during transformation.

**Nucleotide sequencing of gene-specific PCR products.** Genomic DNA was isolated from PCR-confirmed transformed and control soybean plants. The fatty acid desaturase 2 (FAD2) gene was amplified using gene-specific primers, and PCR products were purified and verified on 1.2% agarose gel. Purified amplicons were outsourced for

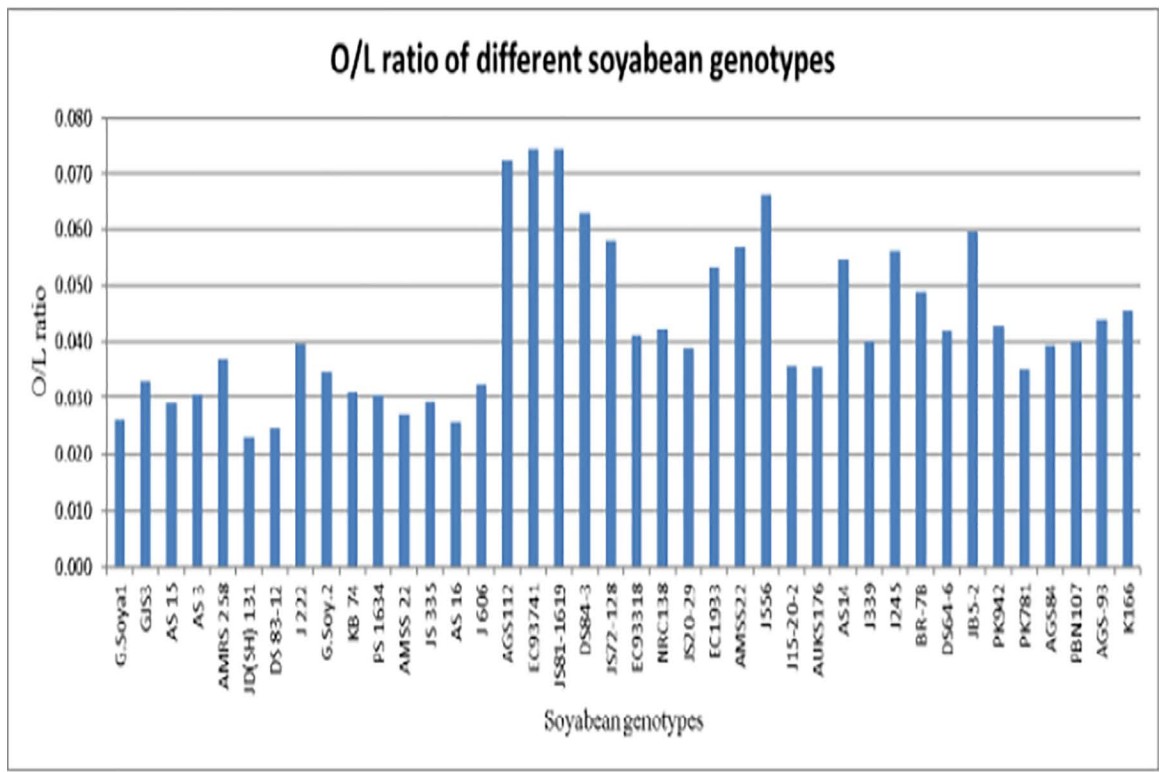

**Fig 6. Total oil content (%) among 40 Indian soybean genotypes.** Bars represent mean values of three replicates.

**Table 4. Effect of 2,4-D and BAP concentrations on callus regeneration from cotyledon explants of soybean cv. GJS-3 (CRD, n = 3).**

| Sr. No. | Media Code | 2,4-D (mg L⁻¹) | BAP (mg L⁻¹) | Callus regeneration (%) |
|---|---|---|---|---|
| 1 | SDB-0 | 0.0 | 0.0 | 14.05[d] |
| 2 | SDB-1 | 0.1 | 0.1 | 42.67[c] |
| 3 | SDB-2 | 0.5 | 0.5 | 53.00[bc] |
| 4 | SDB-3 | 1.0 | 1.0 | 57.00[bc] |
| 5 | SDB-4 | 2.0 | 2.0 | 73.67[b] |
| 6 | SDB-5 | 3.0 | 3.0 | 84.33[a] |
| 7 | SDB-6 | 3.5 | 3.5 | 79.33[ab] |

Error bars represent standard error (SE) of three biological replicates (n = 3). One-way ANOVA showed significant genotype effects for all fatty acids (p < 0.001).

nucleotide sequencing (AgriGenome Labs Pvt. Ltd.). Sequence quality was assessed using BioEdit software, which revealed clear chromatogram peaks with minimal background noise, confirming high-quality sequencing data.

**Bioinformatic analysis of targeted mutations.** Sequencing of the *FAD2* target region identified site-specific mutations in three plants (T3, T7, and T15), resulting in an overall editing frequency of 13.63% (3/22). When calculated relative to PCR-positive plants, the editing efficiency among transformants was 21.4% (3/14). Nucleotide sequences were aligned with the reference FAD2 gene using BioEdit. Analysis revealed targeted mutations in three putative transformed lines of soybean cv. GJS-3. All three lines exhibited substitution-type mutations located near the PAM sequence and within

**Table 5. Effect of BAP concentrations on shoot regeneration from cotyledon explants of soybean cv. GJS-3 (CRD, n = 3).**

| Sr. No. | Media Code | BAP (mg L$^{-1}$) | Shoot regeneration (%) |
|---|---|---|---|
| 1 | SB-0 | 0.0 | 18.62[f] |
| 2 | SB-1 | 0.5 | 46.33[e] |
| 3 | SB-2 | 1.0 | 52.33[de] |
| 4 | SB-3 | 2.0 | 65.00[cd] |
| 5 | SB-4 | 2.5 | 75.67[bc] |
| 6 | SB-5 | 3.0 | 85.33[a] |
| 7 | SB-6 | 3.5 | 79.00[ab] |

Error bars represent standard error (SE, n = 3). One-way ANOVA indicated significant variation among genotypes ($F_{39,80}$ = 28.73, p < 0.001).

**Table 6. Effect of BAP and GA$_3$ concentrations on shoot multiplication and elongation from cotyledon explants of soybean cv. GJS-3 (CRD, n = 3).**

| Sr. No. | Media Code | BAP (mg L$^{-1}$) | GA (mg L$^{-1}$) | Shoot multiplication and elongation (%) |
|---|---|---|---|---|
| 1 | SBG-0 | 0.0 | 0.0 | 26.09[g] |
| 2 | SBG-1 | 0.5 | 0.1 | 45.33[f] |
| 3 | SBG-2 | 1.0 | 0.2 | 61.67[e] |
| 4 | SBG-3 | 1.5 | 0.3 | 67.00[de] |
| 5 | SBG-4 | 2.0 | 0.4 | 79.00[cd] |
| 6 | SBG-5 | 2.5 | 0.5 | 81.67[bc] |
| 7 | SBG-6 | 3.0 | 1.0 | 87.67[a] |

Error bars represent standard error (SE, n = 3). One-way ANOVA revealed highly significant variation ($F_{39,80}$ = 67.45, p < 0.001).

**Table 7. Effect of various concentrations of IBA on root regeneration from multiple shoot explants in Soybean cv. GJS-3.**

| Sr. No. | Media Code | IBA (mg L$^{-1}$) | Root induction and multiplication (%) |
|---|---|---|---|
| 1 | SI-0 | 0.0 | 11.05[f] |
| 2 | SI-1 | 0.1 | 49.67[e] |
| 3 | SI-2 | 0.5 | 50.33[e] |
| 4 | SI-3 | 1.0 | 54.00[e] |
| 5 | SI-4 | 1.5 | 69.67[d] |
| 6 | SI-5 | 2.0 | 86.33[a] |
| 7 | SI-6 | 2.5 | 78.33[b] |

One-way ANOVA confirmed highly significant genotype effects for all fatty acids (p < 0.001), with F-values ranging from 34.56 (linolenic acid) to 61.29 (linoleic acid), indicating substantial genetic diversity in fatty acid biosynthesis among the tested cultivars.

the distal end of the gRNA target site, confirming CRISPR/Cas9-mediated double-strand breaks and repair (Figs 10–11). The overall mutation frequency was 13.63%. This frequency was calculated as three edited lines out of 22 regenerated plants, with mutations (two T→C, one A→C) predicted to impair FAD2 activity and thereby elevate oleic acid levels compared with the wild type. Specifically, thymine-to-cytosine substitutions were observed in two lines (T3 and T7), while adenine-to-cytosine substitution occurred in one line (T15).

**Estimated economic benefits of elevated oleic acid.** Oleic acid content in edited GJS-3 lines was increased relative to wild-type (Table 1). Using a conservative market premium of US$0.10 per lb for identity-preserved high-oleic oil (based on 2022 North American market data; range US$0.05–0.25 per lb) and an illustrative oil yield of 200 kg oil·ha$^{-1}$,

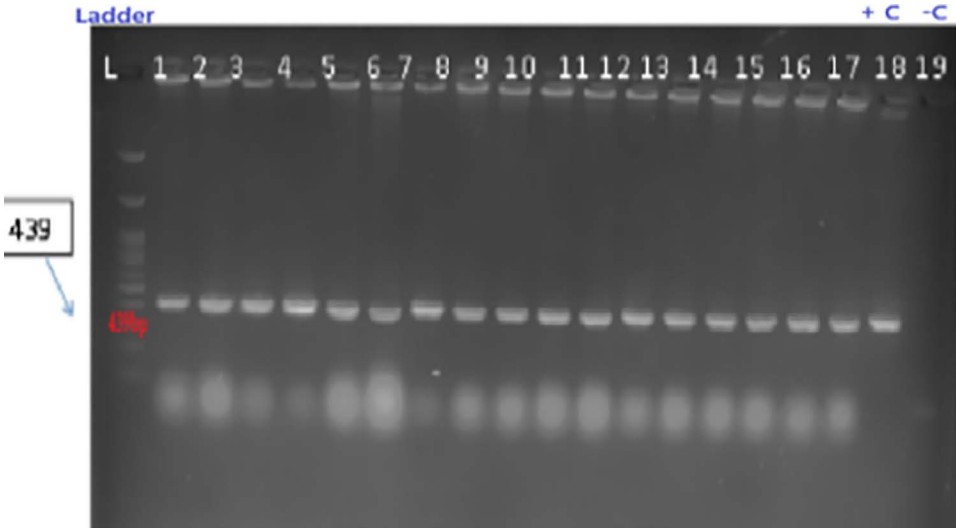

**Fig 7. Oleic-to-linoleic acid (O/L) ratio among 40 Indian soybean genotypes.** Higher O/L ratio indicates superior oil stability.

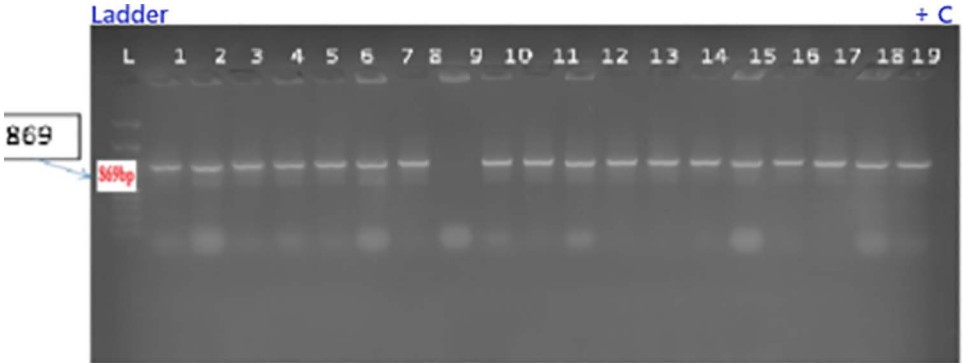

**Fig 8. Colony PCR analysis for presence of Cas9A (439 bp) in E. coli DH5α. L: 1 kb ladder.** Lanes 1–17: Transformed colonies. Lane 18: Positive control. Lane 12: Negative control.

the estimated gross farmer benefit is ≈ US$40/ha (Supplementary S10 Table in S2 File). Processing-cost avoidance from eliminating partial hydrogenation was estimated at US$15.6–79.0 per tonne of oil (low-mid-high scenarios) based on component-level analysis. At representative oil yields (200 kg/ha), this translates to processing savings of US$3.12–15.80 per hectare (Supplementary S11 Table in S2 File). Note that Indian domestic markets currently lack established premium structures for high-oleic soybean varieties; these estimates represent potential export value or future domestic premiums contingent on market infrastructure development and quality certification systems.

## Discussion

### Enhancing soybean oil quality through genetic modification

Soybean oil historically faced market challenges due to high linolenic acid causing undesirable taste [14,15]. While early breeding and hydrogenation approaches were effective [16–18], they suffered from 8–12-year cycles, linkage drag, and limited natural FAD2 variation. CRISPR/Cas9 overcomes these limitations through direct elite cultivar modification without

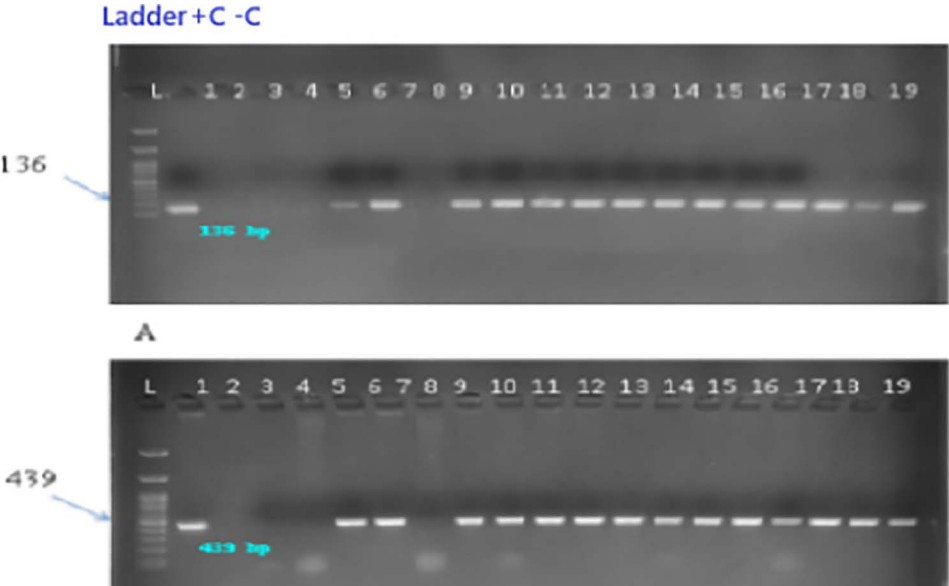

**Fig 9. Colony PCR analysis for presence of Cas9B (~690 bp) in Agrobacterium tumefaciens LBA4404.** L: 1 kb DNA ladder. Lanes 1–18: Transformed colonies. Lane 19: Positive control.

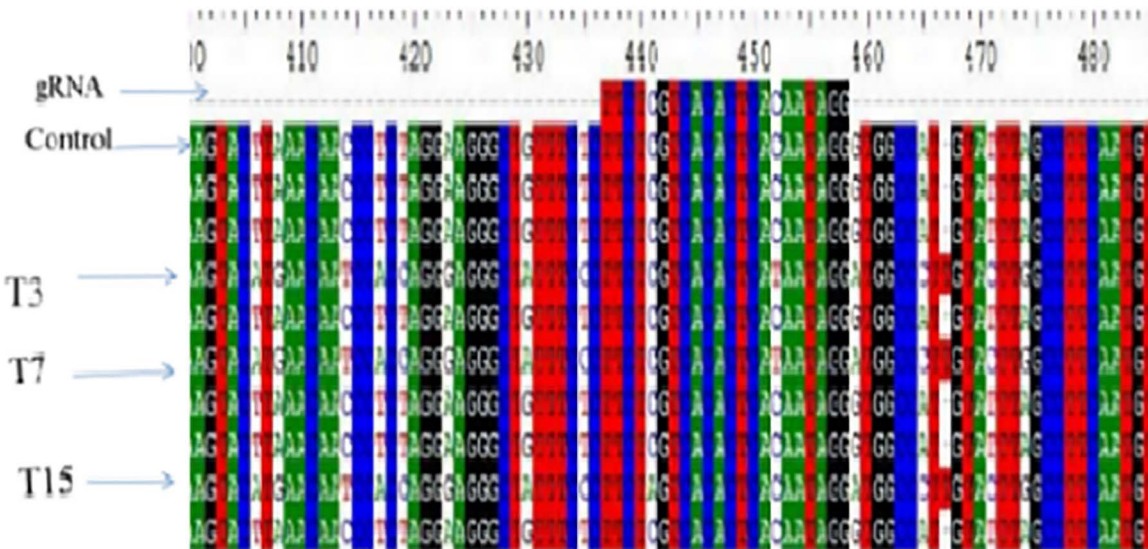

**Fig 10. PCR confirmation of transgene integration in putative transgenic soybean plants. (a)** U3 promoter amplification (136 bp). **(b)** Cas9A fragment amplification (439 bp). L: 1 kb DNA ladder. Lane 1: Positive control. Lane 2: Negative control. Lanes 6, 7, 19: Transformed lines.

linkage drag, reducing development time to 3–5 years and enabling precise allele engineering unavailable through natural variation [19]. This technological advance holds significant promise for targeted oil quality improvement.

Although selective breeding has been effective for desired plant traits, its reliance on random recombination is often inefficient and faces substantial limitations for high-oleic trait development: 8–12-year breeding cycles, linkage drag

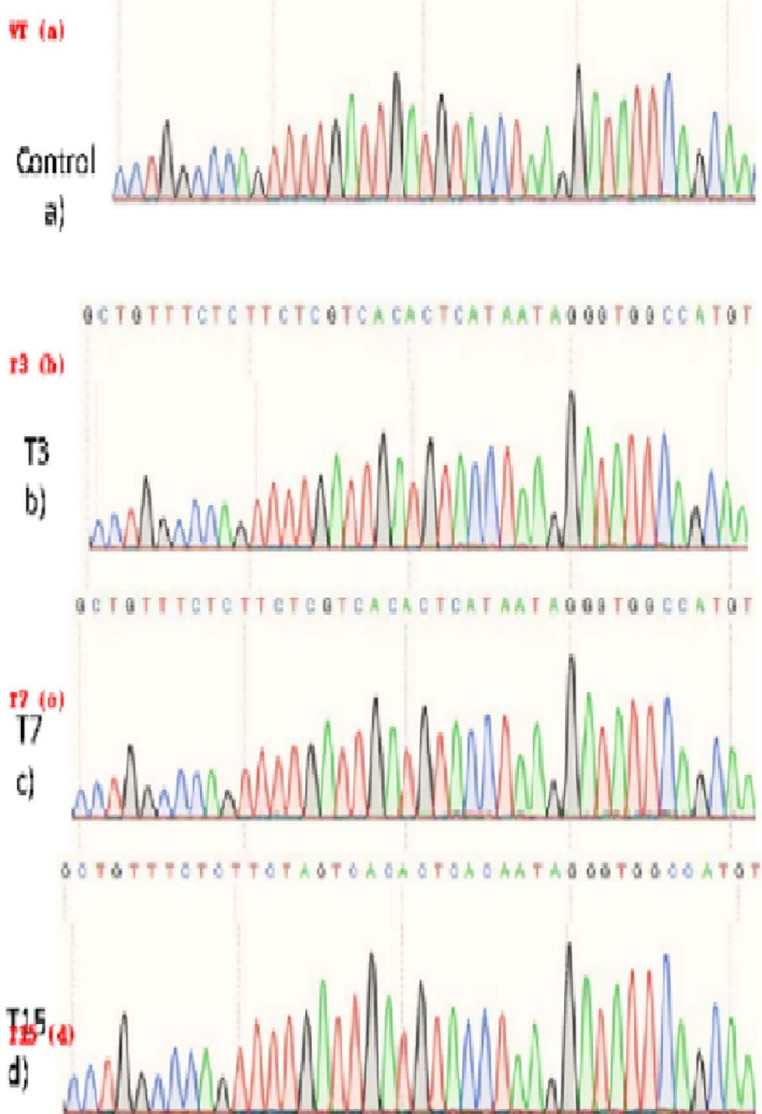

**Fig 11. Sanger sequencing chromatograms of wild-type GJS-3 (control) and three edited soybean lines (T3, T7, T15). Red arrows indicate nucleotide substitutions near the PAM site.**

during introgression requiring extensive backcrossing, and limited natural genetic variation for FAD2 mutations. These constraints are particularly problematic when developing varieties that combine high oleic acid with regional adaptation, disease resistance, and yield potential. However, recent advancements in genetics, particularly the CRISPR/Cas9 system, offer precise and effective plant genome editing that circumvents these limitations by enabling direct modification in elite cultivars without linkage drag, reducing development time to 3–5 years, and allowing precise allele engineering unavailable through natural variation. These advantages complement and extend earlier genome editing methods such as ZFNs and TALENs [19]. This technological leap holds significant promise for targeted soybean modifications, aiming to overcome the high linolenic acid challenge and boost marketability.

The selection of Indian soybean germplasm addresses a key translational gap in genome-editing research. Indian cultivars differ in fatty acid profiles from North American high-oleic lines due to distinct breeding histories, agro-climatic adaptation, and end-use priorities, with oil quality being particularly critical for human consumption in India. High temperatures during seed filling may further influence fatty acid desaturation dynamics, potentially altering the response to FAD2 disruption. These region-specific factors highlight the need for empirical validation of genome-editing strategies in Indian genetic backgrounds. Our work with GJS-3 therefore provides the first demonstration of effective CRISPR/Cas9-mediated FAD2 editing in an Indian soybean cultivar, supporting the development of locally adapted high-oleic varieties.

## Unlocking the potential of soybean for superior oil quality

Our analysis of 40 soybean genotypes revealed significant genetic diversity in fatty acid composition, with the consistent presence of five major fatty acids (palmitic, stearic, oleic, linoleic, and linolenic) demonstrating inherent variation valuable for oil quality improvement [20–24]. Total oil content ranged from 16.89% to 19.79% [25], with G. Soya 2 representing a valuable high-yielding parent line. The correlation between oil content and fatty acid composition suggests that yield and quality improvements can be achieved concurrently [26,27]. Considerable variation in oleic-to-linoleic (O/L) ratios among genotypes provides a powerful indicator for targeted genetic modifications [28]. Cultivars with high O/L ratios (EC9374, JS81–1619) are prime candidates for breeding superior oxidative stability, while low O/L genotypes emphasize the need for meticulous selection [29]. This variability, consistent with cultivar, accession type, and ecoregion-dependent profiles reported across oilseed crops [30–32], provides strategic foundation for developing varieties meeting economic and nutritional demands.

## Genetic modification for superior soybean traits

Our study successfully regenerated plantlets from cotyledonary node explants, demonstrating the effectiveness of our optimized tissue culture protocols. We observed high rates of shoot multiplication and elongation using specific concentrations of BAP and GA, indicating these growth regulators are vital for enhancing regeneration efficiency [33,34]. The ability to generate multiple shoots from a single explant is incredibly beneficial for rapid propagation and genetic transformation, directly aiding the development of transgenic lines with improved oil quality [35].

A foundational principle in plant biotechnology is that successful genetic transformation requires a reliable *in vitro* regeneration protocol. Both direct and indirect genetic transformation methods, especially those using *Agrobacterium tumefaciens*, hold immense promise for improving major crop species [36–40]. In this study, we specifically focused on *in vitro* regeneration and genetic transformation of Soybean (GJS-3) using cotyledonary explants. We meticulously selected elite, healthy explants from germinated seeds, acknowledging that explant age significantly impacts regeneration success [41]. This precise approach is crucial for effectively utilizing biotechnological tools in crop improvement.

## Targeted gene editing in soybean: A key to enhanced oil quality

In the present study, *E. coli* DH5α and *Agrobacterium tumefaciens* LBA4404 were used for cloning and transformation, respectively. The Cas9-sgRNA construct was initially transformed and cloned in *E. coli*, then subsequently introduced into Agrobacterium. Successful plasmid transfer was confirmed through colony PCR, leveraging the inherent virulence of *Agrobacterium tumefaciens* strain LBA4404 for efficient transformation in both cotyledonary node and half-seed explants.

Our protocol maximized transformation efficiency by pre-culturing wounded explants for three days prior to co-cultivation with Agrobacterium. This critical step promotes the accumulation of chemicals that stimulate the vir genes in the Agrobacterium plasmid [42]. Standardizing the co-cultivation duration with *Agrobacterium tumefaciens* is crucial for effective gene transfer. We found that optimal regeneration was achieved after 30 minutes of inoculation and three days of co-cultivation, carefully avoiding excessive bacterial proliferation and subsequent tissue necrosis [43,44]. This aligns with previous findings where three-day co-cultivation periods significantly increased soybean cotyledonary node transformation rates, with both shorter and longer durations proving less effective.

The integration of the Cas9-gRNA construct into the *in vitro* regenerated transformed plants was confirmed through PCR amplification of the construct's components, using specific primers and subsequent electrophoretic gel visualization. Our PCR results revealed that 57.1% of the transformed plants successfully integrated all CRISPR/Cas9 components, indicating a promising rate of gene editing system integration [45] Importantly, this also means that 42.8% of the transformed plants did not integrate the Cas9 gene during T-DNA transfer.

The increase in oleic acid and concomitant reduction in linoleic acid observed in the edited GJS-3 lines can be mechanistically attributed to disruption of *FAD2* function. *FAD2* encodes an endoplasmic reticulum–localized Δ12 fatty acid desaturase responsible for converting oleic acid (18:1) to linoleic acid (18:2) during seed lipid biosynthesis. Single-nucleotide substitutions within the *FAD2* coding region likely impair enzyme activity by affecting conserved residues involved in substrate binding or catalysis, thereby reducing desaturation flux. Consequently, oleic acid accumulates while linoleic acid levels decline. The moderate oleic acid increase, relative to near-complete enrichment reported in multiplex *FAD2* knockouts, is consistent with partial loss-of-function mutations and the heterozygous nature of the $T_0$ edits. These results support a dosage-dependent role of *FAD2* in fatty acid composition and demonstrate that single-locus disruption can significantly improve oil quality in Indian soybean germplasm. This precision in gene editing is crucial for developing high-oleic soybean lines. By disrupting target pathways without introducing foreign DNA, CRISPR/Cas9 potentially mitigates regulatory hurdles associated with traditional GMOs. Standardized PCR and Sanger sequencing provided robust molecular validation [46]. Although Agrobacterium-mediated transformation and FAD2's role in oleic acid regulation have been reported in North American and East Asian cultivars [47], our study represents the first CRISPR/Cas9-mediated FAD2 editing in an Indian soybean cultivar. This addresses whether editing strategies developed in temperate germplasm translate to tropical/subtropical genetic backgrounds with distinct fatty acid biosynthesis. Despite lower baseline oleic acid in Indian cultivars, FAD2 disruption achieved substantial proportional increases, confirming functional conservation of this metabolic target across diverse germplasm—essential for regional breeding programs requiring local agro-climatic adaptation. Because FAD2 is expressed in both seeds and vegetative tissues, mutations may have pleiotropic effects on other agronomic traits. Literature from other oilseed crops provides relevant context: in *Arabidopsis*, FAD2 knockout lines showed altered chilling sensitivity and membrane fluidity due to changes in lipid unsaturation, though growth and reproductive fitness remained largely unaffected under optimal conditions [48]. In *Brassica napus*, high-oleic lines generated through FAD2 suppression exhibited normal agronomic performance under field conditions but showed modest reductions in cold germination rates, attributed to altered seed membrane composition [49]. Conversely, cotton plants with reduced FAD2 expression displayed enhanced drought tolerance, potentially due to decreased lipid peroxidation under water stress. [50].

These findings suggest specific priorities for follow-up evaluation of our edited GJS-3 lines: (i) multi-location field trials across diverse Indian agro-climatic zones to assess yield stability, maturity, and stress responses; (ii) targeted evaluation of cold germination and seedling vigor, particularly relevant for winter sowing systems in northern India; (iii) assessment of membrane integrity and oxidative stress tolerance under high-temperature conditions during seed filling; and (iv) comprehensive phenotyping of vegetative traits including flowering time, plant height, lodging resistance, and seed quality parameters (protein content, germination rate, seed size). Our preliminary greenhouse observations indicate normal morphology and development in edited lines, but multi-season field evaluation will be essential to detect subtle phenotypic effects that may emerge under variable environmental conditions.

Furthermore, previous studies have shown that edits at different FAD2 target sites can produce variable effects on oleic acid accumulation [51], suggesting that target-site selection and allele-specific differences may influence oil-quality outcomes.

The successful hardening and acclimatization of *in vitro* regenerated plantlets indicate that the plants can adapt to ex vitro conditions, which is a critical step in the transformation process. The initial establishment of plantlets in potting mixtures demonstrates the viability of the regenerated plants and their potential for further growth and development in

field conditions. This step is essential for ensuring the survival of transformed plants and their subsequent evaluation for agronomic performance. [41]. The high-oleic phenotype confers nutritional and economic advantages. Identity-preserved premiums for high-oleic soybean oil range from US$0.10–0.20 per lb (≈US$200–440 per tonne) [52,53]. Combined with processing savings from eliminating hydrogenation (Supplementary S10–S11 Table in S2 File), edited lines offer value to farmers and processors while reducing trans-fat formation and public health risks.

To contextualize our findings, we benchmark key metrics against published FAD2 editing studies. Our editing efficiency (13.63%) falls within typical ranges [11,47,48] demonstrating effective CRISPR function in Indian germplasm. Our oleic acid enhancement (22%→33–34%) represents substantial improvement, though modest compared to multiplex approaches (Haun: 80% via dual FAD2-1A/1B editing), reflecting our single-locus strategy prioritizing regulatory simplicity. Our transformation rate (57.1%) exceeds typical soybean rates (Jung:～35%; Du:～41%), attributed to optimized co-cultivation protocols. Key innovations include: (i) first FAD2 editing in Indian germplasm, validating translation of temperate-developed strategies to tropical/subtropical backgrounds with distinct fatty acid biosynthesis; (ii) transgene-free confirmation via dual-target PCR; and (iii) high-efficiency regeneration (84–87%) in locally adapted cultivar, providing a breeding-ready framework for regional programs.

An important limitation of this study is that trait heritability has not yet been confirmed across multiple generations. The edited lines reported here are T0 plants carrying heterozygous single-nucleotide substitutions that are expected to segregate in Mendelian ratios. We are currently advancing T1 progeny to evaluate inheritance patterns, phenotypic stability, and agronomic performance. Confirmation of stable heritability across T1/T2 generations will be essential for breeding applications and regulatory approval.

Additionally, while PCR analysis confirms absence of detectable Cas9 and U3 sequences in edited lines, providing evidence for transgene-free status, we acknowledge that PCR cannot definitively rule out small fragmented T-DNA integrations or copy number variations. However, we performed PCR amplification targeting two independent regions of the T-DNA construct (Cas9: 439 bp; U3: 136 bp), both showing complete absence of amplification in edited lines while producing strong signals in positive controls. This dual-target approach, combined with uniform phenotypes consistent with clean segregation, provides robust evidence against T-DNA integration. Recent CRISPR soybean studies reported transgene-free status based on PCR validation alone, reflecting current field standards [12,48]. For lines advancing to regulatory submission, comprehensive molecular characterization including Southern blot analysis or whole-genome sequencing will be performed to provide definitive confirmation, consistent with regulatory requirements for genome-edited crop varieties.

Although our sgRNA exhibited high in silico specificity (CHOPCHOP score 97.6, zero predicted off-target sites with ≤4 mismatches), experimental validation through targeted sequencing or whole-genome sequencing was not performed, representing a study limitation. The stringent target selection criteria (≤3 seed-region mismatches, ≤4 total mismatches) minimize off-target risk, but definitive confirmation requires experimental validation. For lines advancing to commercial release, comprehensive off-target analysis through targeted amplicon sequencing or whole-genome sequencing will be conducted, consistent with regulatory standards for genome-edited crops.

## Conclusion

This study validates CRISPR-Cas9 efficacy for enhancing soybean oil quality and establishes a robust Agrobacterium-mediated transformation protocol using cotyledonary explants of Indian cultivar GJS-3. By targeting the FAD2 gene, we induced site-specific mutations that elevated oleic acid and reduced linoleic acid content, improving oxidative and thermal stability while reducing hydrogenation requirements. Beyond molecular validation, edited lines offer industrial and economic value through identity-preserved premiums for high-oleic oil, lower processing costs, and reduced trans-fat formation, benefiting farmers, processors, and consumers. Taken together, this represents the first successful application of CRISPR/Cas9-mediated FAD2 editing in an Indian soybean cultivar, demonstrating that editing strategies developed in North American and East Asian germplasm translate effectively to tropical and subtropical genetic backgrounds despite

distinct baseline fatty acid profiles and environmental adaptation patterns. This work offers not only a framework for healthier and more stable oil production but also validates CRISPR/Cas9 as a cost-effective and sustainable strategy for region-specific crop improvement, breeding programs, and germplasm release that can address India's edible oil security challenges through locally adapted high-oleic varieties.

## Supporting information

**S1 File. Supporting Figures S1–S13.**
(PDF)

**S2 File. Supporting Tables S1–S11.**
(PDF)

**S3 File. Figure S14 (Original uncropped gel for Figure 1).**
(JPG)

**S4 File. Figure S15 (Original uncropped gel for Figure 2).**
(JPG)

**S5 File. Figure S16 (Original uncropped gel for Figure 3).**
(JPG)

**S6 File. Figure S17 (Original uncropped gel for Figure 7).**
(JPG)

**S7 File. Figure S18 (Original uncropped gel for Figure 8).**
(JPG)

**S8 File. Figure S19 (Original uncropped gel for Figure 9).**
(JPG)

## Author contributions

**Conceptualization:** Sunil Tulshiram Hajare.

**Data curation:** Balaji U. Rathod, Riddhi Rajyaguru, Ramesh N. Dhawale, Rukam S. Tomar, Shasikant Sharma, Omar Awad Alsaidan.

**Formal analysis:** Balaji U. Rathod, Riddhi Rajyaguru, Ramesh N. Dhawale, Shasikant Sharma, Omar Awad Alsaidan.

**Funding acquisition:** Shasikant Sharma.

**Investigation:** Balaji U. Rathod.

**Methodology:** Rukam S. Tomar, Sunil Tulshiram Hajare.

**Resources:** Riddhi Rajyaguru, Ramesh N. Dhawale.

**Software:** Ramesh N. Dhawale.

**Supervision:** Rukam S. Tomar, Sunil Tulshiram Hajare.

**Validation:** Riddhi Rajyaguru, Ramesh N. Dhawale, Manohar G. Chaskar, Omar Awad Alsaidan.

**Visualization:** Balaji U. Rathod, Manohar G. Chaskar, Omar Awad Alsaidan.

**Writing – original draft:** Manohar G. Chaskar, Sunil Tulshiram Hajare.

**Writing – review & editing:** Manohar G. Chaskar, Sunil Tulshiram Hajare.

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
