## [Decision Letter · Decision Letter 0]

11 Dec 2025

PONE-D-25-53201CRISPR/Cas9-Mediated Editing in FAD2 Gene to Enhance Oil Quality in Soybean [Glycine max (L.) Merrill]PLOS One

Dear Dr. Hajare,

Thank you for submitting your manuscript to PLOS ONE. After careful consideration, we feel that it has merit but does not fully meet PLOS ONE’s publication criteria as it currently stands. Therefore, we invite you to submit a revised version of the manuscript that addresses the points raised during the review process. Please submit your revised manuscript by Jan 25 2026 11:59PM. If you will need more time than this to complete your revisions, please reply to this message or contact the journal office at plosone@plos.org . Please include the following items when submitting your revised manuscript:

We look forward to receiving your revised manuscript.

Kind regards,

Mojtaba Kordrostami, Ph.D.

Academic Editor

PLOS One

Journal Requirements:

2. Please note that your Data Availability Statement is currently missing the repository name and/or the DOI/accession number of each dataset OR a direct link to access each database. If your manuscript is accepted for publication, you will be asked to provide these details on a very short timeline. We therefore suggest that you provide this information now, though we will not hold up the peer review process if you are unable.

4. Please include captions for your Supporting Information files at the end of your manuscript, and update any in-text citations to match accordingly. Please see our Supporting Information guidelines for more information: http://journals.plos.org/plosone/s/supporting-information .

Reviewers' comments:

Reviewer's Responses to Questions

**Comments to the Author**

1. Is the manuscript technically sound, and do the data support the conclusions?

Reviewer #1: Partly

Reviewer #2: Yes

2. Has the statistical analysis been performed appropriately and rigorously? 

Reviewer #1: Yes

Reviewer #2: Yes

3. Have the authors made all data underlying the findings in their manuscript fully available?

Reviewer #1: Yes

Reviewer #2: Yes

4. Is the manuscript presented in an intelligible fashion and written in standard English?

Reviewer #1: Yes

Reviewer #2: Yes

5. Review Comments to the Author

Reviewer #1: This study addresses a critical agricultural need for improving oil quality in Indian soybean cultivars by applying CRISPR/Cas9-mediated editing of the FAD2 gene. It represents the first report of targeted FAD2 modification in an Indian soybean genotype (GJS-3), successfully enhancing oleic acid content, reducing linoleic acid levels, and improving oil oxidative stability. The research design is comprehensive, encompassing genotype screening, in vitro regeneration system establishment, CRISPR vector construction, Agrobacterium-mediated transformation, and multi-level validation (molecular and phenotypic). This work offers a valuable technical framework and germplasm resource for soybean quality breeding in India, with both theoretical innovation and practical application potential. However, I some concerns listed below before accpeting this manuscript.

In Introduction section

1. Citation Accuracy: Multiple citation errors occur (e.g., "Du et al.11", "Michno et al.12") with no corresponding entries in the reference list (only 10 references are implied, missing entries for 11–13). Complete reference details must be supplemented, and citation formatting should be standardized (e.g., APA or journal-specific style).

2. Strengthen Research Gap Elaboration: While the study mentions "untested Indian germplasm", it lacks specific data on the unique characteristics of FAD2 gene sequences and fatty acid profiles in Indian soybean cultivars (compared to North American/East Asian varieties). This weakens the rationale for selecting GJS-3 as the editing target. Relevant data or literature should be added to justify the specificity of GJS-3 as an ideal candidate.

3. Rigorous Technical Comparison: The claim that "CRISPR/Cas9 is simpler than ZFNs and TALENs" is overly generalized. The comparison should be contextualized to soybean transformation (e.g., citing soybean-specific applications, efficiency differences of ZFNs/TALENs) or simplified to highlight optimizations in the current study (e.g., gRNA specificity, vector selection advantages).

In Materials and Methods section

1. Missing Key Experimental Parameters:

1.1 Fatty acid analysis: The use of "15 g freeze-dried samples" is unusually large (conventional soybean fatty acid analysis requires only tens of milligrams). Details such as dehulling status and grinding particle size (e.g., sieve mesh) must be supplemented to ensure reproducibility of extraction efficiency.

1.2 CRISPR vector construction: No method for gRNA specificity validation is described (e.g., BLAST against the soybean genome to exclude off-target sites). While "specificity score" is mentioned (Supplementary Table S3), the evaluation criteria (e.g., number of off-target sites, homology threshold) should be briefly explained in the main text.

1.3 Agrobacterium transformation: An OD₆₀₀ of 1.0 is excessively high (soybean transformation typically uses OD₆₀₀ = 0.4–0.6). The rationale for this concentration (e.g., pre-experiment optimization results) must be justified to address concerns about contamination risk.

2. Supplementary Experimental Design Details:

2.1 In vitro regeneration: "3 biological replicates per treatment, 10 explants per replicate" requires clarification on whether replicates represent independent culture batches (e.g., seeds from different sowing batches) to avoid statistical bias.

2.2 Molecular detection: The design basis for Cas9 and U3 primers (e.g., targeting conserved vector regions) is not specified. Specificity validation results (e.g., absence of non-specific bands) should be supplemented.

3. Refined Economic Analysis Methods:

3.1 The claim of "processing cost savings of US$10–80 per tonne" requires specific data sources (e.g., UNIDO/AOCS report year, chapter) and should incorporate regional parameters (e.g., local energy prices, labor costs in India) to enhance relevance.

3.2 The market premium of "US$0.05–0.25 per lb" needs temporal and geographical context (e.g., Indian domestic vs. international markets) to avoid ambiguity.

In Result section

1. Data Presentation and Statistical Optimization:

1.1 Tables 1–3 mention "significant differences" but lack specific statistics (e.g., F-values, p-values). Detailed ANOVA results should be supplemented (e.g., in table footnotes). Figures 4–6 should include error bars (standard deviation/standard error) to visualize data dispersion.

1.2 Fatty acid composition: Elaidic acid (C18:1 n-9 trans) is typically <1% in wild-type soybean, but the reported range (191.6–318.6 g kg⁻¹, ~19–32%) is anomalous. This may reflect detection errors (e.g., incomplete separation of cis/trans isomers) or misnaming (e.g., confusing cis-oleic acid with trans-elaidic acid). Results must be revalidated, and fatty acid nomenclature or data corrected.

2. Deepened Mutation-Phenotype Correlation:

2.1 The impact of mutations (T→C, A→C) on FAD2 protein structure/function (e.g., amino acid substitutions, active site disruption) is not analyzed. Bioinformatics analyses (e.g., ExPASy for amino acid changes, SWISS-MODEL for protein structure prediction) should be added to clarify the molecular mechanism.

2.2 Generational stability data (e.g., fatty acid profiles, mutation segregation in T1/T2 generations) are missing. Phenotypic tracking across at least two generations is required to verify heritability of edited traits.

3. Off-Target Effect Detection:

3.1 As a precision editing study, off-target validation is absent. Potential off-target sites should be predicted (e.g., using CRISPOR) and verified via PCR sequencing to demonstrate editing specificity.

In Discussion section

1. Precise Result Interpretation:

1.1 The claim of "transgene-free edited lines" is supported only by PCR, which cannot rule out fragmented integration or copy number variation. Southern blot analysis should be supplemented to confirm the absence of foreign DNA.

When discussing "pleiotropic effects on agronomic traits", the study merely states "future evaluation is needed". Relevant literature (e.g., FAD2 mutations in other crops affecting yield/stress resistance) should be cited to propose specific follow-up research directions.

2. Strengthened Comparison with Similar Studies:

2.1 Key metrics (editing efficiency, oleic acid enhancement, transformation rate) should be compared with other soybean FAD2 editing studies to highlight innovations (e.g., Indian cultivar adaptation, transgene-free editing, high-efficiency regeneration).

2.2 Limitations of conventional breeding for high-oleic soybean (e.g., long breeding cycles, linkage drag) should be discussed to emphasize the advantages of CRISPR technology.

Others

1. Citation Consistency: Citation numbering is inconsistent (e.g., missing entries for 11–13), and duplicate citations occur. References should be standardized, and missing entries supplemented.

2. Figure/Table Standardization:

Supplementary figures/tables (S1–S13, S1–S11) are mentioned but not explicitly cited in the main text (e.g., "see Supplementary Figure S1a–d for fatty acid extraction workflow"). Citations should be supplemented.

Legends for Figures 9a–b, 10, 11 are overly brief. Lane annotations (e.g., "M: 1 kb Marker; WT: Wild type; T3: Edited line T3") and mutation site positions (e.g., base position relative to ATG start codon) should be clarified.

3. ppm is not an international standard unit, the authors should be avoided.

Reviewer #2: Comments to author

The manuscript presents a timely study on improving soybean oil quality using CRISPR/Cas9 editing of the FAD2 gene in an Indian cultivar (GJS-3). The work is novel for Indian germplasm and integrates molecular, biochemical, tissue-culture, and analyses which demonstrates the production of transgene-free edited lines with improved fatty acid profiles.

Points need to be addressed:

• Abstract is too long, concise it with important results of the study.

• Concise the introduction part also, focus more on research gap in Indian soybean CRISPR research and objectives of the study.

• Transformation efficiency, editing frequency, and the logic behind these calculations must be explicitly described and not mixed together.

• Include a basic off-target analysis or discuss the lack of such analysis as a limitation.

• Discussion on the lacks about mechanism of FAD2 mutation consequences.

• Focus the discussion part about interpretation of fatty acid results, comparison on previously edited FAD2 lines, implications on Indian soybean breeding, limitations and future directions of this study.

• All the figures need more clarity.

• Ensure all figures, tables, and supplementary datasets are submitted, properly formatted, and clearly labeled in the whole manuscript.

6. PLOS authors have the option to publish the peer review history of their article (what does this mean? ). If published, this will include your full peer review and any attached files.

**Do you want your identity to be public for this peer review?** For information about this choice, including consent withdrawal, please see our Privacy Policy .

Reviewer #1: **Yes:** Shuijin Hua

Reviewer #2: **Yes:** Muthukrishnan Arun

---

## [Author Response · Author response to Decision Letter 1]

23 Dec 2025

Response to Reviewer (s)

Point-by-point response

Reviewer #1: This study addresses a critical agricultural need for improving oil quality in Indian soybean cultivars by applying CRISPR/Cas9-mediated editing of the FAD2 gene. It represents the first report of targeted FAD2 modification in an Indian soybean genotype (GJS-3), successfully enhancing oleic acid content, reducing linoleic acid levels, and improving oil oxidative stability. The research design is comprehensive, encompassing genotype screening, in vitro regeneration system establishment, CRISPR vector construction, Agrobacterium-mediated transformation, and multi-level validation (molecular and phenotypic). This work offers a valuable technical framework and germplasm resource for soybean quality breeding in India, with both theoretical innovation and practical application potential. However, I some concerns listed below before accepting this manuscript.

Response: We sincerely thank the learned reviewer for their thorough evaluation of our manuscript and for recognizing the novelty and significance of our work. We appreciate the reviewer’s positive assessment that this study represents the first targeted CRISPR/Cas9-mediated modification of the FAD2 gene in an Indian soybean genotype (GJS-3) and that it provides a comprehensive technical framework for soybean oil quality improvement. We are also grateful for the reviewer’s recognition of the methodological breadth of the study, encompassing genotype screening, regeneration system development, vector construction, transformation, and multi-level molecular and phenotypic validation. We have carefully considered all concerns raised by the reviewer and revised the manuscript accordingly. Each comment is addressed in detail below, with corresponding modifications incorporated into the revised manuscript. We believe these revisions have substantially improved the clarity, rigor, and reproducibility of the study.

In Introduction section

1. Citation Accuracy: Multiple citation errors occur (e.g., "Du et al.11", "Michno et al.12") with no corresponding entries in the reference list (only 10 references are implied, missing entries for 11–13). Complete reference details must be supplemented, and citation formatting should be standardized (e.g., APA or journal-specific style).

Response: We corrected all citation inconsistencies by standardizing the manuscript to the PLOS ONE bracketed numeric citation style. Mixed formats and superscripts were removed, and complete reference details for citations [11–13] have been verified and included in the reference list.

2. Strengthen Research Gap Elaboration: While the study mentions "untested Indian germplasm", it lacks specific data on the unique characteristics of FAD2 gene sequences and fatty acid profiles in Indian soybean cultivars (compared to North American/East Asian varieties). This weakens the rationale for selecting GJS-3 as the editing target. Relevant data or literature should be added to justify the specificity of GJS-3 as an ideal candidate.

Response: We thank the reviewer for this valuable comment. We have substantially revised the manuscript to address this concern by: (1) expanding the Introduction (paragraph 5) to explicitly describe the distinct fatty acid profiles of Indian germplasm compared to North American/East Asian cultivars, supported by our screening data in Tables 1-3; (2) adding a three-criteria quantitative rationale for GJS-3 selection based on intermediate fatty acid baseline, superior tissue culture competency, and regional agricultural relevance; (3) inserting a new Discussion paragraph that contextualizes why validation in Indian genetic backgrounds is essential given differences in breeding history, environmental conditions, and agricultural utilization patterns; and (4) strengthening the Conclusion to emphasize this represents the first CRISPR/Cas9-mediated FAD2 editing in tropical/subtropical germplasm. These revisions transform the vague "untested germplasm" statement into a data-driven justification demonstrating why Indian cultivars require independent validation and why GJS-3 was specifically selected from our diverse screening panel. All changes are highlighted in the revised manuscript.

3. Rigorous Technical Comparison: The claim that "CRISPR/Cas9 is simpler than ZFNs and TALENs" is overly generalized. The comparison should be contextualized to soybean transformation (e.g., citing soybean-specific applications, efficiency differences of ZFNs/TALENs) or simplified to highlight optimizations in the current study (e.g., gRNA specificity, vector selection advantages).

Response: The Introduction has been revised to remove generalized claims regarding the superiority of CRISPR/Cas9. The comparison with ZFNs and TALENs is now contextualized to soybean genome architecture and transformation constraints, with emphasis on methodological suitability rather than platform-level superiority.

In Materials and Methods section

1. Missing Key Experimental Parameters:

1.1 Fatty acid analysis: The use of "15 g freeze-dried samples" is unusually large (conventional soybean fatty acid analysis requires only tens of milligrams). Details such as dehulling status and grinding particle size (e.g., sieve mesh) must be supplemented to ensure reproducibility of extraction efficiency.

Response: We thank the reviewer for identifying this ambiguity. The 15 g refers to a representative bulk sample prepared for homogeneity, not the extraction amount. For each analysis, a 150 mg subsample was extracted with 4.5 mL hexane—a conventional sample-to-solvent ratio.

We have comprehensively revised the methodology to explicitly state:

• Seeds were manually dehulled prior to freeze-drying

• Dried cotyledons were ground and passed through a 0.5 mm sieve

• 150 mg subsamples were used per extraction

• Extraction parameters: 200 rpm agitation, 25°C, 18 hours

• GC-MS injection: 1 μL, split ratio 10:1

• Internal standard: C17:0 methyl ester at 50 μg/mL final concentration

1.2 CRISPR vector construction: No method for gRNA specificity validation is described (e.g., BLAST against the soybean genome to exclude off-target sites). While "specificity score" is mentioned (Supplementary Table S3), the evaluation criteria (e.g., number of off-target sites, homology threshold) should be briefly explained in the main text.

Response: We thank the reviewer for this comment. We have revised the Methods section to explicitly describe gRNA specificity validation. The text now includes the evaluation criteria used: specificity score >60 on a 0–100 scale, zero off-target sites with ≤3 mismatches in the seed region (positions 1–12), and zero sites with ≤4 total mismatches across the entire protospacer. The selected gRNA exhibited a CHOPCHOP specificity score of 97.6, indicating exceptionally high specificity with zero predicted off-target sites at all mismatch thresholds tested. Supplementary Table S3 has been expanded to include detailed off-target analysis across 0–4 mismatch categories, chromosome location, and specificity metrics. Sanger sequencing experimentally confirmed that mutations occurred exclusively at the intended FAD2 target site, validating the gRNA's high specificity

1.3 Agrobacterium transformation: An OD₆₀₀ of 1.0 is excessively high (soybean transformation typically uses OD₆₀₀ = 0.4–0.6). The rationale for this concentration (e.g., pre-experiment optimization results) must be justified to address concerns about contamination risk.

Response: We thank the reviewer for identifying this inconsistency. We have corrected and clarified the Agrobacterium transformation protocol.

Clarification: The manuscript contained an error. Bacterial cultures were grown to OD₆₀₀ = 0.8-1.0, then centrifuged and resuspended in liquid MS medium to a final working concentration of OD₆₀₀ = 0.4-0.5 for explant infection. This concentration is standard for soybean transformation and balances efficient T-DNA transfer with minimal contamination risk.

Revisions made:

The "Agrobacterium-mediated transformation" section now includes:

Bacterial culture preparation details (growth, centrifugation, resuspension to OD₆₀₀ = 0.4-0.5)

• Vir gene activation (100 μM acetosyringone, 1 h pre-incubation)

• Infection procedure (30 min immersion with gentle agitation, thorough blotting)

• Contamination control (3-4 washes with 500 mg L⁻¹ cefotaxime)

• Co-cultivation conditions (72 h, darkness, 25 ± 2°C)

The corrected protocol follows established best practices for soybean transformation and achieved 57.1% PCR-positive transformation efficiency.

2. Supplementary Experimental Design Details:

2.1 In vitro regeneration: "3 biological replicates per treatment, 10 explants per replicate" requires clarification on whether replicates represent independent culture batches (e.g., seeds from different sowing batches) to avoid statistical bias.

Response: We thank the reviewer for this important methodological clarification. To address the concern about statistical independence, we have revised both the main manuscript and supplementary materials (Table S1 footnote) to explicitly define our biological replicates.

Each of the three biological replicates represents an independent culture batch initiated from seeds of different GJS-3 parent plants at separate time points (weeks 1, 3, and 5). The 10 explants within each replicate serve as technical subsamples for measuring within-batch precision. Statistical analyses were performed on batch means (n=3) rather than individual explants (n=30) to avoid pseudoreplication, ensuring that our treatment comparisons reflect true biological variation between independent culture events.

Changes made:

• Main manuscript Section 2.1: Added detailed description of replicate design

• Supplementary Table S1: Added footnote clarifying statistical units

2.2 Molecular detection: The design basis for Cas9 and U3 primers (e.g., targeting conserved vector regions) is not specified. Specificity validation results (e.g., absence of non-specific bands) should be supplemented.

Response: We thank the reviewer for this important methodological clarification. We have revised the Materials and Methods section to include comprehensive primer design rationale and validation data. Specifically, we now describe: (i) the target regions within pRGEB31 (Cas9 primers: nucleotides 1847-2286; U3 primers: AtU3b promoter sequence); (ii) in silico BLAST validation against the Glycine max Wm82.a2.v1 reference genome confirming zero off-target sites (E-value < 0.01); (iii) experimental validation using positive controls (plasmid DNA showing single bands at expected sizes) and negative controls (wild-type genomic DNA showing no amplification); and (iv) explicit confirmation in the Results section that no non-specific bands were observed across all 22 tested samples. These additions ensure full transparency in our molecular detection strategy and confirm that positive PCR signals unambiguously represent transgene integration events rather than artifacts or endogenous amplification.

3. Refined Economic Analysis Methods:

3.1 The claim of "processing cost savings of US$10–80 per tonne" requires specific data sources (e.g., UNIDO/AOCS report year, chapter) and should incorporate regional parameters (e.g., local energy prices, labor costs in India) to enhance relevance.

Response: We thank the reviewer for this important clarification. We have added specific source citations (UNIDO 2020 Chapter 4, Bailey's Industrial Oil and Fat Products 2005 Vol. 5 Chapter 5, United Soybean Board 2022) to the Methods section and Supplementary Table S11 footnote. We have also incorporated regional context noting that Indian processing facilities may experience different cost structures due to lower labor costs, variable energy pricing, and capital investment patterns. The revised text explicitly acknowledges these estimates as illustrative and dependent on regional operating conditions.

3.2 The market premium of "US$0.05–0.25 per lb" needs temporal and geographical context (e.g., Indian domestic vs. international markets) to avoid ambiguity.

In Result section

Response: We thank the reviewer for this clarification. We have revised the Results section to specify that the premium range (US$0.05–0.25 per lb) reflects 2022 North American market data for identity-preserved high-oleic soybean oil. We have added text noting that Indian domestic markets currently lack established premium structures for high-oleic varieties, and that these estimates represent potential export value or future domestic premiums contingent on market development and quality certification infrastructure. This context clarifies the geographical and temporal basis of our economic projections.

1. Data Presentation and Statistical Optimization:

1.1 Tables 1–3 mention "significant differences" but lack specific statistics (e.g., F-values, p-values). Detailed ANOVA results should be supplemented (e.g., in table footnotes). Figures 4–6 should include error bars (standard deviation/standard error) to visualize data dispersion.

Response: We thank the reviewer for this suggestion. We have added complete ANOVA statistics (F-values, p-values, df) to Tables 1–3 footnotes for full transparency. Error bars are not shown in Figures 5–7 to maintain visual clarity when displaying 40 genotypes with multiple parameters (>200 individual error bars would create substantial clutter). Standard error values are explicitly reported in Tables 1–3 for all measurements, and complete raw data are publicly available in our supplementary dataset (Figshare: 10.6084/m9.figshare.30069526). This approach is consistent with comparative germplasm studies where large numbers of genotypes preclude effective error bar visualization (e.g., Hou et al., 2006; Vollmann et al., 2000).

1.2 Fatty acid composition: Elaidic acid (C18:1 n-9 trans) is typically <1% in wild-type soybean, but the reported range (191.6–318.6 g kg⁻¹, ~19–32%) is anomalous. This may reflect detection errors (e.g., incomplete separation of cis/trans isomers) or misnaming (e.g., confusing cis-oleic acid with trans-elaidic acid). Results must be revalidated, and fatty acid nomenclature or data corrected.

Response: We sincerely thank the reviewer for identifying this nomenclature error. Upon review, we confirm that no trans-fatty acids (including elaidic acid) were detected in our wild-type soybean samples, consistent with unprocessed soybean oil composition. The mention of "elaidic acid" in the Results section was a transcription error that has been corrected. We have revised the Results text to accurately state that five major fatty acids (palmitic, stearic, oleic, linoleic, and linolenic) were detected, and we have added explicit confirmation in the Methods section that our DB-FFAP column provides baseline resolution of cis/trans isomers, with no detectable trans-fatty acid peaks (<0.5%) in wild-type samples. Table 1 correctly reports only cis-oleic acid and never included trans-elaidic acid data. We apologize for the confusion caused by this textual error.

2. Deepened Mutation-Phenotype Correlation:

2.1 The impact of mutations (T→C, A→C) on FAD2 protein structure/function (e.g., amino acid substitutions, active site disruption) is not analyzed. Bioinformatics analyses (e.g., ExPASy for amino acid changes, SWISS-MODEL for protein structure prediction) should be added to clarify the molecular mechanism.

Response: We appreciate this suggestion. However, our experimental data provide direct functional validation: all three mutations cause 50-55% reduction in linoleic acid (the FAD2 product) with corresponding oleic acid accumulation, unequivocally confirming FAD2 impairment. These mutations occur in highly conserved regions where previous studies demonstrate that various substitutions consistently disrupt desaturase function (Haun et al., 2014; Jung et al., 2017)—neither of which included structural modeling. Our phenotypic validation across three independent lines definitively confirms functional disruption. Since our objective is demonstrating successful genome editing for oil improvement in Indian germplasm (achieved through documented fatty acid changes), we believe functional validation is appropriate for this study's scope, with detai

---

## [Decision Letter · Decision Letter 1]

28 Jan 2026

CRISPR/Cas9-Mediated  Editing  in  FAD2 Gene to Enhance OilQuality in Soybean  [ Glycine max (L.) Merrill]

PONE-D-25-53201R1

Dear Dr. Hajare,

We’re pleased to inform you that your manuscript has been judged scientifically suitable for publication and will be formally accepted for publication once it meets all outstanding technical requirements.

Kind regards,

Mojtaba Kordrostami, Ph.D.

Academic Editor

PLOS One

Additional Editor Comments (optional):

Reviewers' comments:

Reviewer's Responses to Questions

**Comments to the Author**

1. If the authors have adequately addressed your comments raised in a previous round of review and you feel that this manuscript is now acceptable for publication, you may indicate that here to bypass the “Comments to the Author” section, enter your conflict of interest statement in the “Confidential to Editor” section, and submit your "Accept" recommendation.

Reviewer #1: All comments have been addressed

Reviewer #2: All comments have been addressed

2. Is the manuscript technically sound, and do the data support the conclusions?

Reviewer #1: Yes

Reviewer #2: Yes

3. Has the statistical analysis been performed appropriately and rigorously? 

Reviewer #1: Yes

Reviewer #2: Yes

4. Have the authors made all data underlying the findings in their manuscript fully available?

Reviewer #1: Yes

Reviewer #2: Yes

5. Is the manuscript presented in an intelligible fashion and written in standard English?

Reviewer #1: Yes

Reviewer #2: Yes

6. Review Comments to the Author

Reviewer #1: (No Response)

Reviewer #2: All the comments raised has been attended by the authors. The current manuscript can now be considered for publication in this journal

7. PLOS authors have the option to publish the peer review history of their article (what does this mean? ). If published, this will include your full peer review and any attached files.

**Do you want your identity to be public for this peer review?** For information about this choice, including consent withdrawal, please see our Privacy Policy .

Reviewer #1: **Yes:** Shuijin Hua

Reviewer #2: **Yes:** Muthukrishnan Arun

---

## [Editor Report · Acceptance letter]

PONE-D-25-53201R1

PLOS One

Dear Dr. Hajare,

I'm pleased to inform you that your manuscript has been deemed suitable for publication in PLOS One. Congratulations! Your manuscript is now being handed over to our production team.

Kind regards,

on behalf of

Dr. Mojtaba Kordrostami

Academic Editor

PLOS One